# Why Go Full? Elevating Federated Learning Through Partial Network Updates

**Haolin Wang**$^{\diamond\dagger}$, **Xuefeng Liu**$^{\diamond\heartsuit}$, **Jianwei Niu**$^{\diamond\heartsuit\ddagger}$, **Wenkai Guo**$^{\diamond\dagger}$, **Shaojie Tang**$^{\spadesuit}$

$\diamond$ State Key Laboratory of Virtual Reality Technology and Systems,
School of Computer Science and Engineering, Beihang University, Beijing, China
$\spadesuit$ Center for AI Business Innovation, Department of Management Science and Systems,
University at Buffalo, Buffalo, New York, USA.
$\heartsuit$ Zhongguancun Laboratory, Beijing, China
{wanghaolin, liu_xuefeng, niujianwei, kyeguo}@buaa.edu.cn
shaojiet@buffalo.edu

## Abstract

Federated learning is a distributed machine learning paradigm designed to protect user data privacy, which has been successfully implemented across various scenarios. In traditional federated learning, the entire parameter set of local models is updated and averaged in each training round. Although this full network update method maximizes knowledge acquisition and sharing for each model layer, it prevents the layers of the global model from cooperating effectively to complete the tasks of each client, a challenge we refer to as layer mismatch. This mismatch problem recurs after every parameter averaging, consequently slowing down model convergence and degrading overall performance. To address the layer mismatch issue, we introduce the FedPart method, which restricts model updates to either a single layer or a few layers during each communication round. Furthermore, to maintain the efficiency of knowledge acquisition and sharing, we develop several strategies to select trainable layers in each round, including sequential updating and multi-round cycle training. Our theoretical analysis and experimental results show that the FedPart method significantly outperforms traditional full-network update strategies, achieving faster convergence, greater accuracy, and reduced communication and computational overhead.

## 1 Introduction

Federated learning is a machine learning framework that protects data privacy, which has attracted widespread attention from researchers in recent years [McMahan et al., 2017, Kairouz et al., 2019, Li et al., 2019]. In traditional federated learning, after receiving the global model sent by the server, each client uses their local data to update the entire model parameters set for several iterations; then, the server averages the updated models to obtain a new global model and broadcasts it to all clients, starting the next training round.

While this approach has proven effective in many applications [Hard et al., 2018, Rieke et al., 2020], its convergence speed and ultimate performance are often lower than those of centralized schemes [McMahan et al., 2017, Zou et al., 2023], even when data across clients are independently and identically distributed (i.i.d.). This suggests that while full network updates and sharing enrich each model layer with more knowledge, they may also introduce factors that negatively impact final performance. To further investigate the underlying reason, we conduct an experiment to visualize

---

† Equal contribution. ‡ Corresponding Author.
The source code is available at: `https://github.com/FLAIR-Community/Fling`

38th Conference on Neural Information Processing Systems (NeurIPS 2024).

the update step sizes during each iteration. Typically, in centralized learning, the update step sizes of the model show a downward trend, indicating that the model is gradually converging. However, in federated learning, as shown in Fig. 1a, the update step sizes significantly increase after each parameter averaging. This suggests that after averaging, the gradients calculated by subsequent layers become notably large, indicating inadequate cooperation among layers within the global model, a phenomenon we term as *layer mismatch*. The cause of this issue is illustrated in Figure 2a. The middle section of the figure depicts the local models of each client, which have undergone sufficient local training. Within these local models, the layers cooperate effectively, demonstrating **match**. However, upon aggregating the parameters of each layer, the averaged layers may struggle to maintain this cooperation, resulting in **mismatch**. This layer mismatch can result in two key issues: first, it may prevent the global model in federated learning from converging to the optimal point of the global loss function, thereby compromising performance. Second, the persistent mismatch disrupts the federated learning process at the server level, significantly reducing training efficiency.

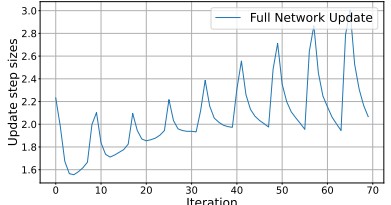

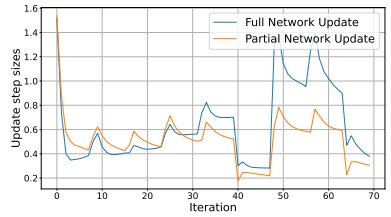

(a) Update step sizes of traditional federated learning with full network updates.

(b) Update step sizes comparison between full network updates and partial network updates.

Figure 1: Update step sizes for each iteration. The experiment uses the ResNet-8 model with 20,000 CIFAR-100 images distributed in an i.i.d. manner across 40 clients.

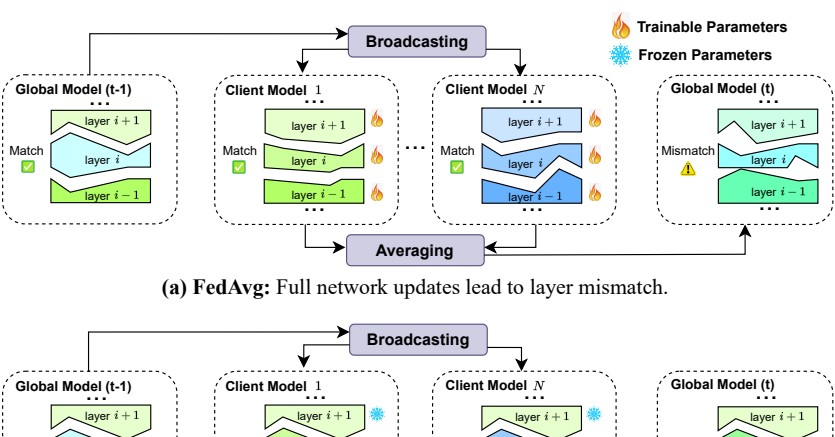

(a) **FedAvg:** Full network updates lead to layer mismatch.

(b) **FedPart:** Partial network updates help to reduce layer mismatch.

Figure 2: Mechanism for layer mismatch in FedAvg and FedPart.

To address the aforementioned problems, we propose FedPart, which employs partial network updates. Our main motivation is illustrated in Fig. 2b. In this toy example, we assume that only the $i$-th layer of the network is trainable in a given round. During local training on each client, this trainable layer can naturally align with the fixed parameters of other layers, which serve as anchors that constrain its update direction. This makes the averaged layers better align with other layers. To validate this approach, we conduct experiments and visualize the results in Figure 1b. The curves demonstrate that partial network updates significantly reduce the increase in update step sizes after averaging, confirming their role in alleviating layer mismatch.

However, as a trade-off, training and transmitting only a portion of the parameters at a time might limit the efficiency of knowledge learning and sharing. Through thorough analysis, we identify a solution to this challenge, which involves carefully selecting trainable parameters. This solution is based on two key principles. The first principle is sequential updating. We train the network layers sequentially, from shallow to deep, one layer at a time. This design is based on the observation that the shallower layers of a neural network typically converge to their final parameters faster than the deeper ones [Raghu et al., 2017]. To align with this natural order, we adopt a similar sequential strategy for layer selection. The second principle is the multi-round cycle training strategy. Our method emphasizes the importance of repeating the process of training from shallow layers to deep layers multiple times. During the original full network updates, shallow layers often learn low-level features, while deep layers learn high-level semantic features [Zeiler and Fergus, 2014, Erhan et al., 2009]. To preserve this property, inspired by the idea of Block Coordinate Descent (BCD) [Poczos and Tibshirani], we propose the multi-round cycling training method to retain this characteristic to the greatest extent.

In addition, FedPart offers improved computation and communication efficiency, making it highly suitable for edge computing scenarios [Wang et al., 2019a,b, Abreha et al., 2022]. This is because FedPart only needs to train a part of the neural network at each training round, thereby significantly reducing the computational overhead of each client in each iteration. At the same time, since clients only need to upload and download the parts of the model that need updating, the amount of parameters to be transmitted is also greatly reduced.

To validate the effectiveness of FedPart, we explore its performance from both theoretical and experimental perspectives. Theoretically, we demonstrate that FedPart has a superior convergence rate under non-convex settings compared to FedAvg. Experimentally, we perform extensive evaluations on various datasets and model architectures. The results indicate that the FedPart method significantly improves convergence speed and final performance (e.g., an improvement of 24.8% on Tiny-ImageNet with ResNet-18), while also reducing both communication overhead (by 85%) and computational overhead (by 27%) simultaneously. Furthermore, our ablation experiments demonstrate how each of the proposed strategies contributes to enhancing the overall performance of FedPart. We also conduct comprehensive visualization experiments to illustrate the underlying rationale of FedPart. In summary, the contributions of this paper are as follows:

- We identify the issue of layer mismatch in federated learning, which arises from updating and aggregating all parameters in each training round. This phenomenon can potentially impact the model's convergence speed and overall performance.

- To mitigate the effects of layer mismatch, we introduce FedPart, which implements partial network updates. Additionally, we develop corresponding strategies for selecting trainable parameters.

- We theoretically analyze the convergence rate of FedPart in a non-convex setting, demonstrating its advantages over full network updates.

- We perform extensive experiments, showing that FedPart achieves significant improvements across multiple evaluation metrics compared to the full network updates scheme. Additionally, ablation and visualization experiments enhance our understanding of the rationale behind FedPart.

## 2   Related Work

Current research on partial parameter training or aggregation in federated learning has led to various applications, broadly categorized into three types:

**Train all parameters, aggregate partial parameters.**   Also known as personalized federated learning, this approach involves each client training all parameters but only aggregating some of them [Tan et al., 2022]. For example, FedPer [Arivazhagan et al., 2019] and FedBN [Li et al., 2021b] personalizes classification and batch-normalization layers respectively, FedRoD [Chen and Chao, 2021] applies both global and local classifier heads. Other works may only upload a low-rank space of parameter matrices [Wang et al., 2023, Wu et al., 2024a]. Although these methods achieve impressive results in data-heterogeneous scenarios, they usually exhibit performance degradation when datasets across clients are distributed in an i.i.d. manner. Moreover, they do not effectively

reduce computational overhead, and any reduction in communication overhead is minimal, as the personalized components are typically small.

**Train partial parameters, aggregate all parameters.** This category refers to each client training a different part of the model, with a global update to the entire model during aggregation. For example, PVT [Yang et al., 2022] and FedPT [Sidahmed et al., 2021] strategically assigns specific model layers to each client, Federated Dropout [Caldas et al., 2018] and FedPMT [Wu et al., 2023] randomly assign a neurons in different layers to clients, HeteroFL [Diao et al., 2020] and FjORD [Horvath et al., 2021] deterministically decide a trainable subnetwork based on client computational power, FedRolex [Alam et al., 2022] further introduces a sliding window method, and CoCoFL [Pfeiffer et al., 2022] introduces a quantization technique for overhead reduction. The primary objective of these methods is to reduce client overhead by dynamically leveraging varying computational capacities among clients. However, compared to full network updates, these methods often result in performance degradation and slower convergence speed.

**Progressive Training.** This approach starts with a small model and gradually increases its size until the entire network is trained [Rusu et al., 2016]. This training paradigm has gain attention in the field of federated learning as its efficiency in reducing resource consumption (e.g., ProgFed [Wang et al., 2022] and ProFL [Wu et al., 2024b]). However, because these methods eventually train a full model, they are not able to solve the layer mismatch problem. Moreover, while these methods aim to reduce resource consumption, they often lead to performance losses compared to full network training. To the best of our knowledge, our FedPart is the first approach to simultaneously enhance both convergence accuracy and efficiency.

## 3 Method

Generally speaking, FedPart is based on partial network updates, which trains and aggregates only a few layers of the global network model for each training round. At the beginning of each training round that requires partial network update, the server first determines which layers need to be trained and sends this information to all clients. Subsequently, each client trains the corresponding layers, transmitting them to the server for aggregation, and the server broadcasts the averaged results to each client for next training round. We elaborate on two key components of FedPart in the following subsections: partial network updates and the strategic selection of trainable layers.

### 3.1 Partial Network Updates

The partial network updates involve training and aggregating only a few layers of the global network model in each later communication rounds. Specifically, we partition the layers of global model into trainable ones and frozen ones. For each training iteration $t$ and client $i$, the optimization objective is: $\arg\min_{w_i^t} \mathbb{E}_{x \sim \mathbf{D_i}}[\mathcal{L}(x|\hat{w}_i^t, \tilde{w}_i^t)]$, where $\hat{w}_i^t$ and $\tilde{w}_i^t$ respectively denotes parameters of trainable and non-trainable layers, $w_i^t \triangleq \{\hat{w}_i^t, \tilde{w}_i^t\}$ represents the total parameter set, $\mathbf{D_i}$ represents the local data distribution of client $i$ and $\mathcal{L}(\cdot)$ refers to the loss function. To optimize this objective, we adopt the following gradient descent formula:

$$w_i^{t+1} = w_i^t - \gamma * S_i^t \odot \nabla_{w_i^t} \mathcal{L}(x|w_i^t), \quad x \sim \mathbf{D_i}. \tag{1}$$

Here, $\gamma$ is the learning rate, $S_i^t$ is a binary mask that selectively enables updates only for trainable parameters and $\odot$ denotes element-wise product. After performing several local training iterations, the parameters of these selected layers are sent to the server and globally averaged at iteration $t = T$: $\bar{w}_T = \frac{1}{N} \sum_{i=1}^{N} w_i^T$, where $N$ represents the number of clients. For the sake of simplicity in formulation, the above equation aggregates and calculates the gradient for all parameters. However, in practical implementation, we only update and transmit the trainable components, significantly reducing both computational and communication costs.

### 3.2 Selecting Trainable Layers

Although training only a subset of parameters can largely mitigate the layer-mismatch issue, it may limit the efficiency of knowledge learning and sharing. Therefore, we propose to carefully select trainable layers to address this limitation. As illustrated in Fig.3, following the initial full network

updates, we train parameters layer by layer from the shallowest to the deepest. Subsequently, we cycle back to the shallowest layer and periodically repeat this process. Generally, this strategy is driven by two key principles:

**Sequential updating.** This principle refers to training model layers in sequence, from shallow to deep layers one at a time. Our motivation is based on the fact that the convergence of neural networks follows a natural intrinsic order, with shallower layers typically converging earlier than deeper ones [Raghu et al., 2017]. By partially updating network in accordance with this inherent training order, we can largely replicate the convergence process of full network updates while preserving training efficiency simultaneously.

**Multi-round cycle training.** This principle refers to repeating the process of updating the neural network layers from shallow to deep multiple times. To illustrate our motivation, consider a fully trained neural network: shallow layers primarily focus on low-level semantic features (such as the edges in images), while deeper layers focus on higher-level semantic features (such as the main objects in images). However, during partial network updates, because the deeper layers are initially non-trainable, shallow layers are forced to learn complex high-level semantic features, which disrupts the original information hierarchy in the neural network. Through multi-round cycle training, we return to the shallow layers after training the deep layers. This strategy can reduce the burden on the shallow layers and helps approximate the final results of full network updates.

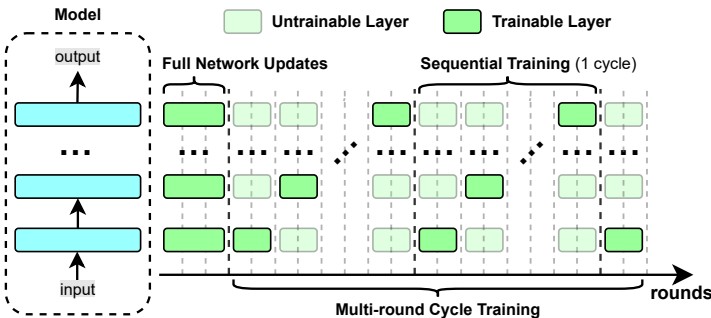

Figure 3: Strategy for selecting trainable layers.

### 3.3 Convergence Analysis for FedPart

To analyze the convergence of FedPart, we first introduce some definitions and notations. Let each client use a uniform loss function $\mathcal{L}(x|w)$ with parameters $w$, for the data $x$ to calculate the loss function value, $f_i(w) = \mathbb{E}_{x \sim \mathbf{D_i}}[\mathcal{L}(x|w)]$ which is the expected loss function of client $i$. In this setting, the overall optimization goal of federated learning can be written as the sum of the expected loss functions of each client, that is: $f(w) = \frac{1}{N}\sum_{i=1}^{N} f_i(w)$. Additionally, for notational simplicity, we denote the parameters of the $i$-th client at time $t$ as $w_i^t$, the computed stochastic gradient vector as $G_i^t$, and the average of all client models at time $t$ as $\bar{w}^t$.

To represent partial network updates, we add a binary matrix $S_i^t$ as a mask for each update process of each client. To keep consistency with the methods section, we assume that for each mask $S_i^t$, $\frac{1}{M}$ of the elements are set to 1, and the rest are set to 0. Before proving the convergence of our FedPart under non-convex conditions, we first introduce three assumptions:

**Assumption 1:** The expected loss function of any client is L-smooth, namely:

$$||\nabla f_i(w) - \nabla f_i(u)|| \le L||w - u||, \forall i, w, u. \tag{2}$$

**Assumption 2:** The variance and second-order moments of the gradients are bounded, that is:

$$\mathbb{E}_{x \sim \mathbf{D_i}}[||\nabla\mathcal{L}(x|w) - \nabla f_i(w)||^2] \le \sigma^2, \forall i, w, x \in \mathbf{D_i},$$
$$\mathbb{E}_{x \sim \mathbf{D_i}}[||\nabla\mathcal{L}(x|w)||^2] \le G^2, \forall i, w, x \in \mathbf{D_i}.$$

**Assumption 3:** The variance of the gradients is approximately equal under all permissible masks:

$$\frac{\mathbb{E}_{x \sim \mathbf{D_i}}[||S_1 \odot (\nabla \mathcal{L}(x|w) - \nabla f_i(w))||]}{\mathbb{E}_{x \sim \mathbf{D_i}}[||S_2 \odot (\nabla \mathcal{L}(x|w) - \nabla f_i(w))||]} \leq k, \forall i, w, x \in \mathbf{D_i}, S_1, S_2. \tag{3}$$

The first two assumptions are common in the literature, ensuring certain necessary characteristics of the loss function. The third assumption ensures that the specific choice of the mask matrix does not have too much impact on the final update variance, provided that the mask meets the required criteria. A further discussion about the third assumption can be found in Appendix G.

Based on these assumptions, we can analyze the convergence of FedPart. In terms of the approximate convergence rate, we align with related literature [Alistarh et al., 2017, Lian et al., 2017, Ghadimi and Lan, 2013], using the average magnitude of the expected gradient over iterations, and finally obtain the following theorem:

**Theorem 1:** Under assumptions 1-3, with the total number of clients as $N$, and all parameters divided into $M$ groups of trainable parameters, the convergence rate of FedPart satisfies:

$$\frac{1}{T} \sum_{t=1}^{T} \mathbb{E}[||S_i^t \odot \nabla f(\bar{w}^{t-1})||^2] = O(\frac{1}{\sqrt{MNT}}), \tag{4}$$

where $\odot$ denotes element-wise product. A detailed proof of this theorem is provided in Appendix B. The results show that compared to the convergence rate of full network updates $O(\frac{1}{\sqrt{NT}})$ [Yu et al., 2019], FedPart demonstrates significantly better convergence performance. This advantage becomes more pronounced as the choice amount of partial parameters per instance is reduced, which aligns with our original intention to reduce layer mismatch. However, it should be noted that convergence analysis can only indicate the difficulty of converging to a stationary point, and cannot measure the model's performance after convergence. Therefore, it is not advisable to arbitrarily reduce the number of parameters trained in each iteration.

### 3.4 Analysis for Communication and Computational Cost

**Communication Cost.** Suppose FedPart divides all layers into $M$ groups, and only one group is trained during each partial training session, with one communication round for each group. It is easy to verify that the averaged communication costs of FedPart during the partial network update phase is reduced to $\frac{1}{M}$ of the original costs.

**Computational Cost.** Assuming that computational costs are uniform across all layers, our method can reduce the overall computational expense during the partial network update phase by $\frac{1}{3}$. The primary reason for this reduction is that in FedPart, there is no need to compute gradients for the layers preceding the trainable parameters. To analyze this quantitatively, suppose the total overhead for both forward and backward propagation in a complete model satisfies $D_{tot} = D_{for} + D_{bak}$. The ratio of computational costs over a single training cycle can then be written as

$$\frac{\boldsymbol{Comp_{PNU}}}{Comp_{FNU}} = \frac{M * D_{for} + \sum_{i=1}^{M} \frac{i*D_{bak}}{M}}{M * D_{for} + M * D_{bak}} = \frac{M * D_{for} + \sum_{i=1}^{M} \frac{(M+1)*D_{bak}}{2M}}{M * D_{for} + M * D_{bak}}.$$

Moreover, it is widely accepted in the literature that the computational cost of backward propagation is approximately twice that of forward propagation [Rasley et al., 2020, Hobbhahn and Sevilla, 2021]. Therefore, the above equation can be rewritten as:

$$\frac{\boldsymbol{Comp_{PNU}}}{Comp_{FNU}} \approx \frac{M * D_{for} + (M+1) * D_{for}}{M * D_{for} + 2M * D_{for}} \approx \frac{2}{3} \tag{5}$$

## 4 Experiments

In the experimental setup, we primarily choose 40 clients, with local epochs to be 8. We test the global model on a balanced set. Unless specifically stated otherwise, the training datasets across all clients are independently and identically distributed (i.i.d.). We utilize the Adam optimizer [Kingma and Ba, 2014] with a learning rate of 0.001, which is determined to be the optimal learning rate. In line

with prior references [Li et al., 2021b, Chen et al., 2022], we refrain from uploading local statistical information during model aggregation. Each experiment is conducted three times with different random seeds to ensure robustness. The experimental results for additional scenarios, including learning-rate tuning and client sampling, are presented in Appendix F.

When choosing experimental metrics, we employ three distinct measures to capture various aspects of the benefits. These metrics include: *Best Acc.*, which represents the ultimate accuracy achieved in classification tasks; *Comm.*, indicating the total upstream transmission volume required by each client for a given training round (in GB); and *Comp.*, which illustrates the total floating-point computation required by each client (in TFLOPs). All experiments are conducted on a server equipped with 8×A100 GPUs, and we provide the complete source code in our supplementary material.

## 4.1 Main Properties

**Comparison with full network updates.** We apply the FedPart method to three classic federated learning algorithms: FedAvg [McMahan et al., 2017], FedProx [Sahu et al., 2018], and FedMOON [Li et al., 2021a], and compare the results with their full network updates (FNU) counterparts. We utilize ResNet-8 [He et al., 2016] (detailed in Appendix A) and update only one layer in each two consecutive training rounds (denoted as 2 R/L). Additionally, we insert five rounds of full network training between each cycle in our FedPart. We conduct experiments on the CIFAR-10 [Krizhevsky et al., 2010], CIFAR-100 [Krizhevsky et al., 2009], and TinyImageNet [Le and Yang, 2015] datasets.

Table 1: Performance of FL algorithms with full network and partial network updates.

| Data | C | FedAvg | | FedProx | | FedMoon | | Comm. | | Comp. | |
|---|---|---|---|---|---|---|---|---|---|---|---|
| | | FNU | FedPart | FNU | FedPart | FNU | FedPart | FNU | FedPart | FNU | FedPart |
| **CIFAR-10** | 1 | 56.0 (±1.1) | 57.7 (±0.5) | 54.4 (±2.1) | 57.5 (±0.6) | 58.9 (±0.5) | 57.8 (±0.4) | 4.83 | 1.35 | 4.38 | 3.21 |
| | 2 | 58.6 (±1.6) | 60.2 (±0.4) | 60.2 (±1.5) | 59.9 (±0.5) | 61.1 (±0.1) | 59.4 (±0.2) | 9.65 | 2.70 | 8.76 | 6.43 |
| | 3 | 59.6 (±1.7) | 61.7 (±0.3) | 62.3 (±0.7) | 61.3 (±0.1) | 62.3 (±0.4) | 59.8 (±0.1) | 14.5 | 4.05 | 13.2 | 9.64 |
| | 4 | 60.7 (±1.3) | **62.8 (±0.2)** | **62.8 (±1.1)** | 62.3 (±0.1) | **62.3 (±0.4)** | 60.5 (±0.6) | 19.3 | **5.40** | 17.5 | **12.9** |
| **CIFAR-100** | 1 | 30.9 (±0.4) | 31.0 (±0.5) | 30.6 (±0.3) | 30.9 (±0.5) | 31.0 (±0.5) | 30.9 (±0.4) | 4.92 | 1.38 | 4.39 | 3.22 |
| | 2 | 32.9 (±0.3) | 34.8 (±0.5) | 33.6 (±0.5) | 34.7 (±0.4) | 33.2 (±0.9) | 35.1 (±0.4) | 9.65 | 2.75 | 8.78 | 6.44 |
| | 3 | 34.3 (±0.2) | 36.1 (±0.5) | 34.5 (±0.5) | 36.7 (±0.4) | 34.6 (±1.1) | 36.5 (±0.6) | 14.8 | 4.13 | 13.2 | 9.66 |
| | 4 | 35.6 (±0.3) | 37.0 (±0.6) | 35.8 (±0.2) | 37.1 (±0.4) | 35.0 (±1.0) | 37.2 (±0.6) | 19.7 | 5.51 | 17.6 | 12.9 |
| | 5 | 35.6 (±0.3) | **37.2 (±0.7)** | 36.2 (±0.5) | **37.5 (±0.2)** | 35.4 (±0.8) | **37.6 (±0.5)** | 24.6 | **6.88** | 21.9 | **16.1** |
| **Tiny-ImageNet** | 1 | 15.6 (±0.6) | 17.1 (±0.2) | 15.8 (±0.4) | 16.8 (±0.2) | 17.5 (±0.6) | 17.3 (±0.3) | 5.02 | 1.40 | 17.5 | 12.9 |
| | 2 | 17.0 (±0.8) | 20.3 (±0.1) | 17.2 (±1.0) | 20.1 (±0.2) | 17.5 (±0.6) | 20.5 (±0.0) | 10.0 | 2.81 | 35.1 | 25.7 |
| | 3 | 17.6 (±0.4) | 20.8 (±0.2) | 18.0 (±0.5) | 20.7 (±0.1) | 18.4 (±0.8) | 21.1 (±0.1) | 15.1 | 4.21 | 52.6 | 38.6 |
| | 4 | 17.7 (±0.4) | 21.1 (±0.1) | 18.2 (±0.7) | 21.2 (±0.1) | 18.4 (±0.8) | 21.5 (±0.1) | 20.1 | 5.62 | 70.1 | 51.4 |
| | 5 | 17.7 (±0.4) | **21.4 (±0.2)** | 18.4 (±0.8) | **21.5 (±0.2)** | 18.4 (±0.8) | **21.7 (±0.1)** | 25.1 | **7.02** | 87.7 | **64.3** |

The results in Table 1 show that our FedPart method demonstrates rapid convergence and consistently outperforms traditional FNU methods across all training cycles **C**, ultimately achieving significantly higher accuracy (e.g., improving FedAvg on Tiny-ImageNet by 21%). At the same time, its communication and computational costs are only 28% and 73% of those required by FNU. Furthermore, we observe that in some scenarios, the performance improvements of other federated learning algorithms even surpass those observed with FedAvg. This highlights that the layer mismatch problem identified in this paper is novel and cannot be addressed by any existing methods. However, our results on CIFAR-10 are less impressive. This suggests that in simpler datasets, the primary issue might be the client drift problem explored in previous studies, whereas the layer mismatch problem becomes more prominent in complex datasets.

**FedPart with deeper models.** To evaluate the effectiveness of FedPart with deeper networks, we conduct experiments on ResNet-18 (detailed in Appendix A). This presents a more challenging scenario, as the proportion of trainable parameters significantly decreases in each round. Our experimental setup also follows the 2 R/L pattern, with five additional full network updates inserted between cycles. The results, displayed in Table 2, show that in deeper networks, FedPart not only maintains its advantages in convergence speed and accuracy but also provides even greater reductions in communication and computational costs (by 85% and 27% compared to full network updates).

Table 2: Performance of FedPart for ResNet-18.

| Data | C | FedAvg-FNU | | | FedAvg-FedPart | | |
|---|---|---|---|---|---|---|---|
| | | Best Acc. | Comm. | Comp. | Best Acc. | Comm. | Comp. |
| CIFAR-10 | 1 | 59.4 (±1.5) | 82.1 | 11.2 | 53.5 (±0.5) | 12.2 | 8.19 |
| | 2 | 61.4 (±0.1) | 164 | 22.3 | 57.5 (±0.6) | 24.5 | 16.4 |
| | 3 | **61.7 (±0.2)** | 246 | 33.5 | 59.2 (±0.4) | **36.7** | **24.6** |
| CIFAR-100 | 1 | 30.4 (±0.4) | 82.5 | 11.2 | 27.8 (±0.5) | 12.3 | 8.20 |
| | 2 | 31.9 (±0.6) | 165 | 22.4 | 31.6 (±0.4) | 24.6 | 16.4 |
| | 3 | 32.0 (±0.5) | 247 | 33.5 | **33.4 (±0.4)** | **36.8** | **24.6** |
| Tiny-ImageNet | 1 | 13.7 (±0.2) | 82.8 | 44.7 | 12.0 (±0.2) | 12.3 | 32.8 |
| | 2 | 13.7 (±0.2) | 166 | 89.4 | 15.1 (±0.3) | 24.7 | 65.5 |
| | 3 | 13.7 (±0.2) | 248 | 134 | **17.1 (±0.2)** | **37.0** | **98.3** |

**FedPart for language modality.** We also extend the FedPart method to the field of natural language processing and evaluate it on AGnews and SogouNews [Zhang et al., 2015] datasets. We choose the transformer architecture [Vaswani et al., 2017] for experiments. As shown in Table 3, the results indicate that FedPart performs well on language tasks, not only maintaining comparable performance as FNU, but also reducing communication and computational overhead by 66% and 25%, respectively. This demonstrates the method's scalability.

Table 3: Performance of FedPart on NLP datasets.

| Data | C | FedAvg-FNU | | | FedAvg-FedPart | | |
|---|---|---|---|---|---|---|---|
| | | Best Acc. | Comm. | Comp. | Best Acc. | Comm. | Comp. |
| AG News | 1 | 91.4 (±0.3) | 22.3 | 5.58 | 91.1 (±0.2) | 7.43 | 4.16 |
| | 3 | 92.0 (±0.2) | 66.9 | 16.7 | 91.5 (±0.2) | 22.3 | 12.5 |
| | 5 | **92.1 (±0.3)** | 106 | 27.9 | 92.0 (±0.3) | **37.2** | **20.8** |
| Sogou News | 1 | 94.2 (±0.2) | 51.8 | 5.58 | 93.8 (±0.2) | 17.3 | 4.16 |
| | 3 | 94.3 (±0.2) | 155 | 16.7 | 94.3 (±0.2) | 51.8 | 12.5 |
| | 5 | **94.4 (±0.2)** | 259 | 27.9 | **94.4 (±0.2)** | 86.3 | **20.8** |

**FedPart under data heterogeneity.** We also evaluate the performance of FedPart under scenarios involving data heterogeneity. The results in Table 4 show that our FedPart consistently improve final performance (e.g., an improvement of 3.4% on Tiny-ImageNet) in the presence of data heterogeneity. However, the extent of performance improvement is relatively smaller. This suggests that client drift [Karimireddy et al., 2020] may have a more pronounced negative impact on our method. We conduct experiments with extreme data heterogeneity ($\alpha = 0.1$) in Appendix F.3.

Table 4: Performance of FedPart under data heterogeneity (Dirichlet, $\alpha = 1$).

| Dataset | C | FedAvg-FNU | FedPart |
|---|---|---|---|
| CIFAR-10 | 2 | 57.7 (± 0.7) | 57.8 (± 0.4) |
| | 3 | 59.2 (± 0.7) | 59.2 (± 0.4) |
| | 4 | 60.4 (± 1.1) | 60.7 (± 0.4) |
| | 5 | 60.4 (± 1.1) | **61.4 (± 0.4)** |
| CIFAR-100 | 2 | 33.1 (± 0.4) | 34.4 (± 0.1) |
| | 3 | 34.3 (± 0.6) | 35.8 (± 0.2) |
| | 4 | 34.9 (± 0.6) | 36.8 (± 0.1) |
| | 5 | 35.2 (± 0.5) | **37.4 (± 0.1)** |
| Tiny-ImageNet | 2 | 16.9 (± 0.3) | 19.8 (± 0.4) |
| | 3 | 17.4 (± 0.1) | 20.3 (± 0.1) |
| | 4 | 17.4 (± 0.1) | 20.4 (± 0.1) |
| | 5 | 17.4 (± 0.1) | **20.8 (± 0.3)** |

Table 5: Performance of FedPart with different training rounds per layer.

| Dataset | R/L | r=15 | r=25 | r=35 | r=45 | r=55 | r=65 |
|---|---|---|---|---|---|---|---|
| CIFAR-10 | 1 | 58.06 | 59.35 | 60.06 | **60.56** | **61.12** | 61.21 |
| | 2 | 56.85 | 58.80 | 58.80 | 60.46 | 60.46 | **61.25** |
| | 4 | 56.17 | 58.76 | 59.60 | 59.60 | 59.60 | 59.60 |
| | 10 | 48.22 | 54.65 | 57.40 | 57.40 | 59.03 | 59.03 |
| CIFAR-100 | 1 | 28.10 | 29.86 | 31.25 | 32.17 | **32.60** | 33.09 |
| | 2 | 24.47 | 30.07 | 30.07 | **32.53** | 32.53 | **33.59** |
| | 4 | 23.56 | 26.26 | 28.19 | 32.01 | 32.01 | 32.01 |
| | 10 | 22.83 | 23.51 | 26.04 | 26.43 | 29.21 | 30.94 |
| Tiny-ImageNet | 1 | 14.37 | 16.33 | 18.02 | **19.27** | **19.88** | 20.18 |
| | 2 | 11.32 | 16.00 | 16.00 | 19.25 | 19.25 | **20.69** |
| | 4 | 9.09 | 11.44 | 15.21 | 17.89 | 17.89 | 17.89 |
| | 10 | 11.33 | 12.03 | 12.03 | 12.03 | 12.03 | 16.16 |

## 4.2 Ablation Study

**Training rounds per layer.** In our FedPart, the training rounds per layer (denoted as R/L) is an important hyperparameter. A larger R/L value means more thorough training in each cycle, but it also results in a decrease in the number of cycles within the same number of training rounds. We explore the performance of FedPart under different R/L. From the results in Table 5, when R/L=1, the outcome shows limited final performance due to insufficient training for each layer. However, further increasing the R/L value not only fails to improve the final performance but also

reduces the convergence speed. In extreme cases, when R/L=10, only one cycle is conducted overall, which significantly affects both convergence speed and final accuracy. This indicates that generally, increasing the number of cycles is more effective than extending their duration. This aligns with the motivation behind our proposal of multi-cycling training.

**Rounds of initial warm-up updates.** To explore the impact of the duration of the initial full network updates phase (i.e. warm-up stage), we conduct experiments with this stage set to lengths of 0, 5, and 60. In Table 6, the term state refers to the period before or after partial network updates, which follow the warm-up phase. The experimental results clearly show that initial full network updates is crucial to the final model's accuracy. Notably, extending the full network update phase yields diminishing returns. However, even when the model is trained with FNU until no further accuracy improvement is observed (60 init.), utilizing FedPart still enhances the model's accuracy. This confirms FedPart's capability to improve the convergence of the final global model and reduce layer mismatch.

Table 6: Impact of the warm-up rounds.

| Dataset | State | 0 init. | 5 init. | 60 init. |
|---|---|---|---|---|
| **CIFAR-10** | bef. | 0 | 41.56 | 58.92 |
| | aft. | 58.48 | 61.25 | 66.18 |
| **CIFAR-100** | bef. | 0 | 20.38 | 34.16 |
| | aft. | 29.53 | 33.59 | 36.65 |
| **Tiny-ImageNet** | bef. | 0 | 9.11 | 16.25 |
| | aft. | 16.81 | 20.69 | 19.99 |

Table 7: Impact of training sequences.

| Dataset | C | Seq. | Rev. | Ran. |
|---|---|---|---|---|
| **CIFAR-10** | 1 | 58.80 | 58.53 | 59.62 |
| | 2 | 60.46 | 59.76 | 59.97 |
| | 3 | **61.25** | 60.19 | 60.23 |
| **CIFAR-100** | 1 | 30.07 | 27.84 | 29.58 |
| | 2 | 32.53 | 29.41 | 30.92 |
| | 3 | **33.59** | 31.79 | 31.44 |
| **Tiny-ImageNet** | 1 | 16.00 | 13.15 | 15.91 |
| | 2 | 19.25 | 15.62 | 17.71 |
| | 3 | **20.69** | 18.33 | 18.99 |

**Different orders for selecting trainable layers.** We experiment with three different orders for selecting trainable parameters: sequential, reverse, and random. Sequential is the default configuration of FedPart, selecting layers from shallow to deep. In contrast, the reverse sequence selects layers from deep to shallow, while the random sequence selects layers randomly in each round. The results of the experiments are depicted in Table 7, demonstrating that the effectiveness of the three methods ranks as follows: *sequential* > *reverse* > *random*. This aligns with the intrinsic convergence order of neural networks and meets our experimental expectations.

## 4.3 Visualization Results

In this section, we conduct experiments to demonstrate why our proposed parameter selection strategy can enhance final performance, and what the impact of layer-wise information exchange has on privacy leakage. Our experiments are based on ResNet-8 and the CIFAR-100 dataset. We analyze the models obtained from four different methods: 1) FedAvg-100, which represents training with full network for 100 rounds; 2) FedPart(No Init. 1C), which represents using FedPart for one cycle without initial full network updates; 3) FedPart(1C), which involves initial full network updates followed by one cycle of FedPart training; 4) FedPart(5C), which involves initial full network updates followed by five cycles of FedPart training. The visualization results are as follows.

**Activation maximization visualization.** Activation maximization [Erhan et al., 2009] involves finding an input that maximizes a specific activation value within a neural network, reflecting the feature patterns the neuron focuses on. We use this method to explore the visual patterns captured by different models and measure their similarity using SSIM (Structural Similarity Index Measure) [Hore and Ziou, 2010]. The results in Table 8 show that, without initial full network updates and multiple cycles, the features captured by the FedPart model significantly differ from those of the FedAvg model. However, this discrepancy decreases after applying our layer-selection strategy, suggesting that the model better recognizes the hierarchical nature of different semantic information, thus enhancing its performance. Additional visual results are provided in Appendix C.

**Convolutional kernel visualization.** We also analyze how different models extract semantic information by visualizing the convolutional kernels. We find that in the full network updates represented by FedAvg-100, the shallow convolutional kernels primarily function as edge/corner detectors. However, direct training of partial networks disrupts this property. Further, by initially employing full network updates and adding multiple training cycles, we gradually restore this characteristic. This effectively explains the impact of the parameter selection strategy on the final model formation. For specific visualization results of the convolutional kernels, please refer to Appendix D.

## 4.4 Impact on Privacy Protection

We next demonstrate that FedPart offers enhanced privacy protection, as it transmits less information in each communication round. Formally, we can abstract the model training process (for both full and partial parameter training) as a mapping: $(\Delta w_1, \Delta w_2, ..., \Delta w_n) = f(x)$, where the left hand side denotes the updates to each model parameter, and $x$ is the training data. From a privacy attack perspective, the goal is to find the best $x$ such that the updates to $w$ are as close as possible to the actual updates in each dimension. This resembles solving a system of equations, where $x$ are the unknowns, and each dimension of $w$ update represents an equation. With partial network training, the unknowns $x$ remain unchanged compared with full parameter training, but the number of equations decreases (i.e., less information for the attacker to leverage). Therefore, we believe that partial network training generally leaks less information.

To verify this experimentally, we conduct several rounds of federated learning using both full network and partial network updates. We employ DLG (Deep Leakage from Gradients) [Zhu et al., 2019] to attempt the recovery of original images and use PSNR (Peak Signal-to-Noise Ratio) [Hore and Ziou, 2010] to measure the extent of privacy leakage. DLG is a classic privacy leakage scheme, which aims at finding an input that produces gradients most similar to the gradients calculated from a given sample. In this way, DLG can approximately recover the input sample. Let the original model input be $x$, then the specific formula for recovering the input $\hat{x}$ is as:

$$\min_{\hat{x}} ||\nabla_{\hat{x}}\mathcal{L}(\hat{x}|w) - \nabla_x\mathcal{L}(x|w)||^2 \tag{6}$$

We use PSNR to measure the quality of the reconstructed image. Given that $x$ denotes the original image and $\hat{x}$ denotes the reconstructed image, the PSNR is calculated as follows:

$$\text{PSNR} = -10 \cdot \log_{10}(\text{MSE}(x, \hat{x})) \tag{7}$$

where $\text{MSE}(x, \hat{x})$ denotes the mean square error between $m \times n$ matrices $x$ and $\hat{x}$, given by:

$$\text{MSE}(x, \hat{x}) = \frac{1}{m \cdot n} \sum_{i=1}^{m} \sum_{j=1}^{n} (x(i,j) - \hat{x}(i,j))^2 \tag{8}$$

A larger PSNR value means a better quality of the reconstructed image, which further implies a higher risk of privacy leakage.

The results in Table 9 show that, for different trainable layers, our method consistently exhibits better privacy protection in both average and worst-case scenarios compared to full network updates. Attacking examples are provided in Appendix E.

Table 8: SSIM of activation maximization images between FedAvg and FedPart.

|  | #1 (Conv) | #10 (FC) |
|---|---|---|
| FedPart(No Init. 1C) | 0.680 | 0.896 |
| FedPart(1C) | 0.863 | 0.955 |
| FedPart(5C) | **0.865** | **0.980** |

Table 9: Average and Max PSNRs of reconstructed images for FedAvg and FedPart models.

| Model | Param. | Avg. PSNR | Max PSNR |
|---|---|---|---|
| FedAvg-100 | All | 17.07 | 25.57 |
| FedPart(5C) | #1 (conv) | **12.53** | **15.02** |
|  | #10 (fc) | 13.84 | 16.88 |

## 5 Conclusion and Limitation

We observe that the model averaged in federated learning is not directly applicable to the specific tasks of each client, a situation we refer to as layer mismatch. To address this issue, we propose the FedPart method, which introduces a strategy for selecting and training partial networks. We validate the effectiveness of FedPart both theoretically and experimentally. In future work, we plan to evaluate our method on a wider range of model architectures and apply it to larger-scale datasets to further investigate its effectiveness and scalability.

---

#i represents the i-th layer of the model, with detailed partitioning method is presented in Appendix A.

## Acknowledgement

This work was supported by the National Natural Science Foundation of China under Grants 62372028 and 62372027.

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

# A Implementation Details

In Section 4, we primarily adopt ResNet and language transformer for experiments, whose architectures are illustrated in Fig. 4 and Fig. 5, respectively.

We also demonstrate the detailed partitioning method in our FedPart. Taking ResNet-8 (on the left in Fig. 4) as an example, we divide the trainable parameters of the model into 10 layers, corresponding to the numbers #1-#10. Among these, the trainable parameters of #1-#9 include not only the weights of the convolutional layers but also the weights and biases of the accompanying BN layers after the convolutional layers. The other models follow the same representation method of layer partitioning. During the sequential training phase of the FedPart method, we select one single layer to train in the order of their numbering #$i$.

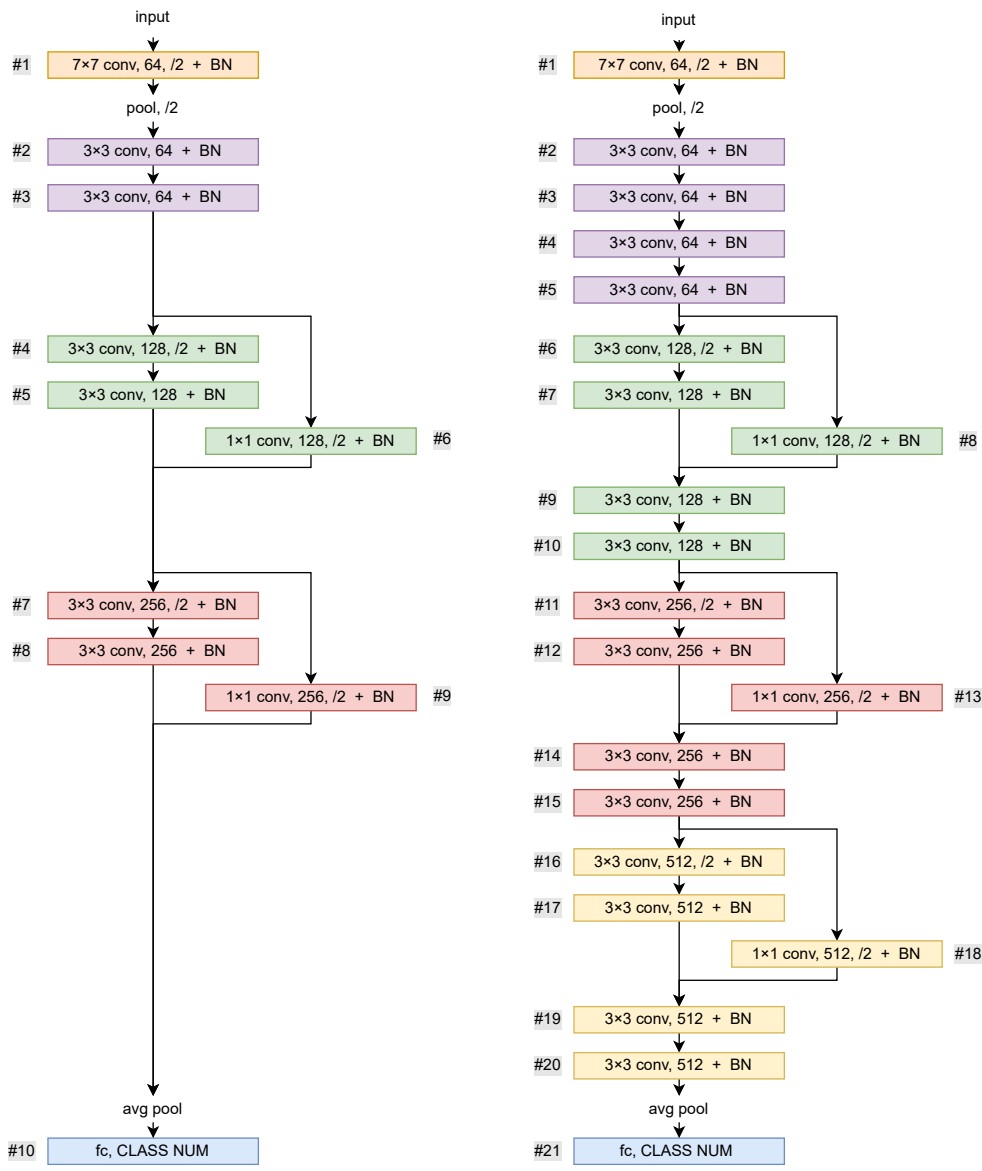

Figure 4: Model architecture and layer partitioning about our ResNet-8 and ResNet-18 model.

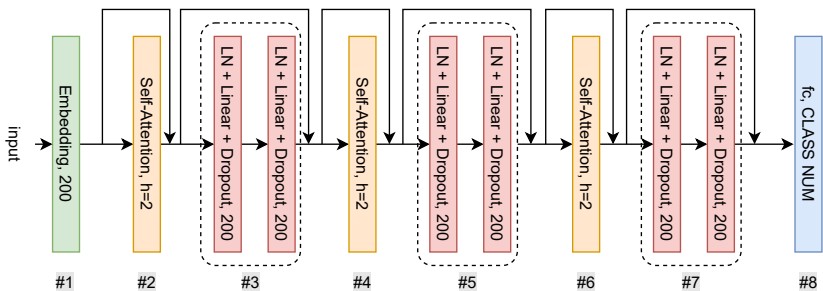

Figure 5: Model architecture and layer partitioning for language transformer.

# B Proof for Convergence Rate of FedPart

Before beginning the proof, we need to analyze the upper bound of the gradient variance after parameter selection. According to Assumption 3, we know that for any mask matrices $S_1, S_2$, it holds that:

$$\frac{\mathbb{E}_{x \sim \mathbf{D_i}}[||S_1 \odot (\nabla \mathcal{L}(x|w) - \nabla f_i(w))||]}{\mathbb{E}_{x \sim \mathbf{D_i}}[||S_2 \odot (\nabla \mathcal{L}(x|w) - \nabla f_i(w))||]} \leq k, \forall i, w, x \in \mathbf{D_i}, S_1, S_2 \tag{9}$$

Constructively, we set a series of mask matrices $S_1, \cdots, S_M$ that have no overlapping '1' elements at the same positions and their sum exactly forms an all-one matrix. Clearly, each of these mask matrices meets our requirements. Therefore, we can derive:

$$\mathbb{E}_{x \sim \mathbf{D_i}}[||S_j \odot (\nabla \mathcal{L}(x|w) - \nabla f_i(w))||^2] \geq \frac{1}{k^2} * \mathbb{E}_{x \sim \mathbf{D_i}}[||S_1 \odot (\nabla \mathcal{L}(x|w) - \nabla f_i(w))||^2] \tag{10}$$

Summing over $j$ from 1 to $M$, the left-hand side of the inequality is exactly the variance of the gradient without any mask matrices. Therefore:

$$\sum_{j=1}^{M} \mathbb{E}_{x \sim \mathbf{D_i}}[||S_j \odot (\nabla \mathcal{L}(x|w) - \nabla f_i(w))||^2] = \mathbb{E}_{x \sim \mathbf{D_i}}[||\nabla \mathcal{L}(x|w) - \nabla f_i(w)||^2]$$

$$\geq \frac{M}{k^2} * \mathbb{E}_{x \sim \mathbf{D_i}}[||S_1 \odot (\nabla \mathcal{L}(x|w) - \nabla f_i(w))||^2]$$

For following proof, we mainly refer to Yu et al. [2019]. According to Assumption 2, the upper limit of the left side of the inequality is $\sigma^2$, so we finally obtain a general upper bound for the gradient with masks:

$$\mathbb{E}_{x \sim \mathbf{D_i}}[||S \odot (\nabla \mathcal{L}(x|w) - \nabla f_i(w))||^2] \leq \frac{\sigma^2 k^2}{M}, \forall i, w, x \in \mathbf{D_i}, S \tag{11}$$

With the groundwork laid, we are now ready to begin the formal proof process. First, based on Assumption 1, as the loss function is L-smooth, we have:

$$\mathbb{E}[f(\bar{w}^t)] \leq \mathbb{E}[f(\bar{w}^{t-1}) + \mathbb{E}[\langle S_i^t \odot \nabla f(\bar{w}^{t-1}), \bar{w}^t - \bar{w}^{t-1}\rangle] + \frac{L}{2}\mathbb{E}[||\bar{w}^t - \bar{w}^{t-1}||^2] \tag{12}$$

Next, we analyze the third term on the right-hand side of the inequality above to derive the following inequality:

$$\mathbb{E}[||\bar{w}^t - \bar{w}^{t-1}||^2] = \gamma^2 \mathbb{E}[||\frac{1}{N}\sum_{i=1}^{N} S_i^t \odot G_i^t||^2]$$

$$= \gamma^2 \mathbb{E}[||\frac{1}{N}\sum_{i=1}^{N} S_i^t \odot (G_i^t - \nabla f_i(w_i^{t-1}))||^2] + \gamma^2 \mathbb{E}[||\frac{1}{N}\sum_{i=1}^{N} S_i^t \odot \nabla f_i(w_i^{t-1})||^2]$$

$$= \frac{\gamma^2}{N^2} \sum_{i=1}^{N} \mathbb{E}[||S_i^t \odot (G_i^t - \nabla f_i(w_i^{t-1}))||^2] + \gamma^2 \mathbb{E}[||\frac{1}{N}\sum_{i=1}^{N} S_i^t \odot \nabla f_i(w_i^{t-1})||^2]$$

$$\leq \frac{\gamma^2 \sigma^2 k^2}{MN} + \gamma^2 \mathbb{E}[||\frac{1}{N}\sum_{i=1}^{N} S_i^t \odot \nabla f_i(w_i^{t-1})||^2]$$

The last inequality comes from the derived Eq. 11. Next, we analyze the second term on the right-hand side of Eq. 12:

$$\mathbb{E}[\langle S_i^t \odot \nabla f(\bar{w}^{t-1}), \bar{w}^t - \bar{w}^{t-1}\rangle] = -\gamma\mathbb{E}[\langle S_i^t \odot \nabla f(\bar{w}^{t-1}), \frac{1}{N}\sum_{i=1}^N S_i^t \odot G_i^t\rangle]$$

$$= -\gamma\mathbb{E}[\langle S_i^t \odot \nabla f(\bar{w}^{t-1}), \frac{1}{N}\sum_{i=1}^N S_i^t \odot \nabla f_i(w_i^{t-1})\rangle]$$

$$= -\frac{\gamma}{2}\mathbb{E}[||S_i^t \odot \nabla f(\bar{w}^{t-1})||^2 + ||\frac{1}{N}\sum_{i=1}^N S_i^t \odot \nabla f_i(w_i^{t-1})||^2-$$

$$||S_i^t \odot \nabla f(\bar{w}^{t-1}) - \frac{1}{N}\sum_{i=1}^N S_i^t \odot \nabla f_i(w_i^{t-1})||^2]$$

$$\leq -\frac{\gamma}{2}\mathbb{E}[||S_i^t \odot \nabla f(\bar{w}^{t-1})||^2 + ||\frac{1}{N}\sum_{i=1}^N S_i^t \odot \nabla f_i(w_i^{t-1})||^2-$$

$$||\nabla f(\bar{w}^{t-1}) - \frac{1}{N}\sum_{i=1}^N \nabla f_i(w_i^{t-1})||^2]$$

Further expanding the right-hand side of the above inequality, we obtain:

$$\mathbb{E}[||\nabla f(\bar{w}^{t-1}) - \frac{1}{N}\sum_{i=1}^N \nabla f_i(w_i^{t-1})||^2] = \mathbb{E}[||\frac{1}{N}\sum_{i=1}^N \nabla f_i(\bar{w}^{t-1}) - \frac{1}{N}\sum_{i=1}^N \nabla f_i(w_i^{t-1})||^2]$$

$$= \frac{1}{N^2}\mathbb{E}[||\sum_{i=1}^N(\nabla f_i(\bar{w}^{t-1}) - \nabla f_i(w_i^{t-1}))||^2]$$

$$\leq \frac{1}{N}\mathbb{E}[\sum_{i=1}^N ||\nabla f_i(\bar{w}^{t-1}) - \nabla f_i(w_i^{t-1})||^2]$$

$$\leq \frac{L^2}{N}\sum_{i=1}^N \mathbb{E}[||\bar{w}^{t-1} - w_i^{t-1}||^2]$$

In the above derivation, we have used the assumption of L-smoothness and Jensen's inequality. Next, we will continue to estimate the upper limit of this term. Assuming that the last parameter aggregation occurred at time $t = t_0$, and the next aggregation will take place at $t = t_0 + E$, then:

$$\mathbb{E}[||\bar{w}^t - w_i^t||^2] = \mathbb{E}[||\gamma\sum_{\tau=t_0+1}^t \frac{1}{N}\sum_{i=1}^N S_i^\tau \odot G_i^\tau - \gamma\sum_{\tau=t_0+1}^t S_i^\tau \odot G_i^\tau||^2]$$

$$= \gamma^2\mathbb{E}[||\sum_{\tau=t_0+1}^t \frac{1}{N}\sum_{i=1}^N S_i^\tau \odot G_i^\tau - \sum_{\tau=t_0+1}^t S_i^\tau \odot G_i^\tau||^2]$$

$$\leq 2\gamma^2\mathbb{E}[||\sum_{\tau=t_0+1}^t \frac{1}{N}\sum_{i=1}^N S_i^\tau \odot G_i^\tau||^2 + ||\sum_{\tau=t_0+1}^t S_i^\tau \odot G_i^\tau||^2]$$

$$\leq 2(t-t_0)\gamma^2\mathbb{E}[\sum_{\tau=t_0+1}^t ||\frac{1}{N}\sum_{i=1}^N S_i^\tau \odot G_i^\tau||^2 + \sum_{\tau=t_0+1}^t ||S_i^\tau \odot G_i^\tau||^2]$$

$$\leq 2(t-t_0)\gamma^2\mathbb{E}[\sum_{\tau=t_0+1}^t \sum_{i=1}^N \frac{1}{N}||S_i^\tau \odot G_i^\tau||^2 + \sum_{\tau=t_0+1}^t ||S_i^\tau \odot G_i^\tau||^2]$$

$$\leq 4(t-t_0)\gamma^2 G^2 \leq 4E\gamma^2 G^2$$

Substituting all the above inequalities into the right side of Eq. 12, we can finally obtain that when using a learning rate $0 \leq \gamma \leq \frac{1}{L}$, it satisfies:

$$\mathbb{E}[f(\bar{w}^t)] \leq \mathbb{E}[f(\bar{w}^{t-1})] - \frac{\gamma - \gamma^2 L}{2}\mathbb{E}[\|\frac{1}{N}\sum_{i=1}^{N}S_i^t \odot \nabla f_i(w_i^{t-1})\|^2$$

$$- \frac{\gamma}{2}\mathbb{E}[\|S_i^t \odot \nabla f(\bar{w}^{t-1})\|^2 + 2\gamma^3 E^2 G^2 L^2 + \frac{L}{2NM}\gamma^2\sigma^2 k^2$$

$$\leq \mathbb{E}[f(\bar{w}^{t-1})] - \frac{\gamma}{2}\mathbb{E}[\|S_i^t \odot \nabla f(\bar{w}^{t-1})\|^2 + 2\gamma^3 E^2 G^2 L^2 + \frac{L}{2NM}\gamma^2\sigma^2 k^2$$

After rearranging the above inequalities, we obtain

$$\mathbb{E}[\|S_i^t \odot \nabla f(\bar{w}^{t-1})\|^2 \leq \frac{2}{\gamma}(\mathbb{E}[f(\bar{w}^{t-1})] - \mathbb{E}[f(\bar{w}^t)]) + 4\gamma^2 E^2 G^2 L^2 + \frac{L}{NM}\gamma\sigma^2 k^2 \qquad (13)$$

Finally, summing the inequalities from $t = 1, \cdots, T$, and multiplying both sides by $\frac{1}{T}$, we obtain:

$$\frac{1}{T}\sum_{i=1}^{T}\mathbb{E}[\|S_i^t \odot \nabla f(\bar{w}^{t-1})\|^2 \leq \frac{2}{\gamma T}(f(\bar{w}^0) - f^*) + 4\gamma^2 E^2 G^2 L^2 + \frac{L}{NM}\gamma\sigma^2 k^2 \qquad (14)$$

Selecting a learning rate $\gamma = \frac{\sqrt{NM}}{L\sqrt{T}}$, we obtain: $\frac{1}{T}\sum_{i=1}^{T}\mathbb{E}[\|S_i^t \odot \nabla f(\bar{w}^{t-1})\|^2 \leq \frac{2L}{\sqrt{NMT}}(f(\bar{w}^0) - f^*) + \frac{4NME^2 G^2}{T} + \frac{\sigma^2 k^2}{\sqrt{NMT}}$. Furthermore, by choosing $E \leq \frac{T^{1/4}}{(MN)^{3/4}}$, we can derive the following corollary: $\frac{1}{T}\sum_{i=1}^{T}\mathbb{E}[\|S_i^t \odot \nabla f(\bar{w}^{t-1})\|^2 \leq \frac{2L}{\sqrt{NMT}}(f(\bar{w}^0) - f^*) + \frac{4G^2}{\sqrt{MNT}} + \frac{\sigma^2 k^2}{\sqrt{NMT}} = O(\frac{1}{\sqrt{NMT}})$. This proves the convergence rate of FedPart.

## C  Visualizations for Activation Maximization

For better visualizing the semantic information recognized by each layer in the different models, in Fig. 6, we present representative results from the first and last layers of models under four scenarios: FedAvg-100, FedPart(No Init. 1C), FedPart(1C), and FedPart(5C).

From the visualization results, it can be observed that FedAvg-100, due to being a full network update, captures low-level semantic features (such as clear boundaries) in shallow layers, while deeper layers capture complex semantic information. However, the results of FedPart(No Init. 1C) exhibit noticeable differences in color and structural features compared to the full network update. This confirms our belief that partial network updates are detrimental to establishing a hierarchical information extraction approach, resulting in the model converging to possible local minima. Additionally, we observe that by including the initial phase of full network updates and multiple rounds of sequential training, the similarity of semantic information obtained by the model gradually approaches that of FedAvg. Therefore, the results sufficiently demonstrate that although we only train one layer of the network each time, by employing an appropriate layer selection scheme, we ultimately achieve results comparable to those of full network updates.

## D  Visualizations for Convolutional Kernel

To visually depict the characteristics of the convolutional kernels in the first convolutional layer of different models, we conduct kernel visualization. The four models we select come from the following scenarios: FedAvg-100, FedPart(No Init. 1C), FedPart(1C), and FedPart(5C).

In Fig. 7, we present a comparison of results for planes in the first convolutional layer. It can be seen that the kernels in the first convolutional layer of the FedAvg-100 model are mostly edge and corner detectors. In contrast, the results of FedPart(No Init. 1C) and FedPart(1C) appear more random and irregular. However, after training to convergence, the results of FedPart(5C) are noticeably more similar to those of FedAvg-100, and start to exhibit characteristics of simple feature extractors. This indicates that through partial network updates, the layers of the model gradually coordinate with each other, yielding a cooperative effect.

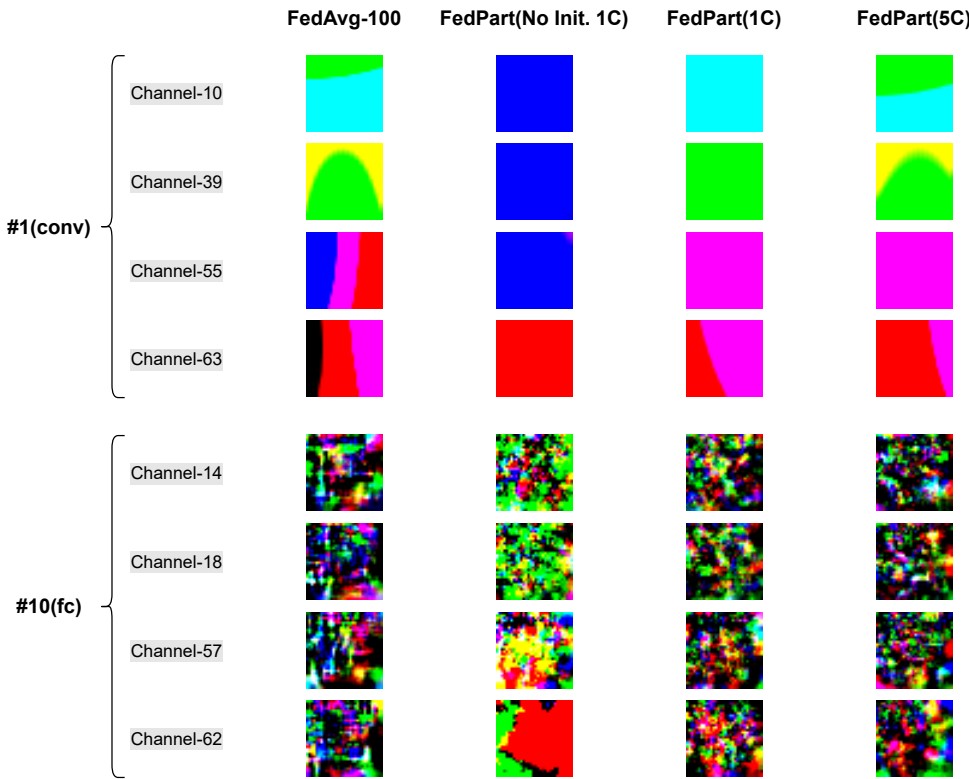

Figure 6: Activation maximization images of different channels within different layers.

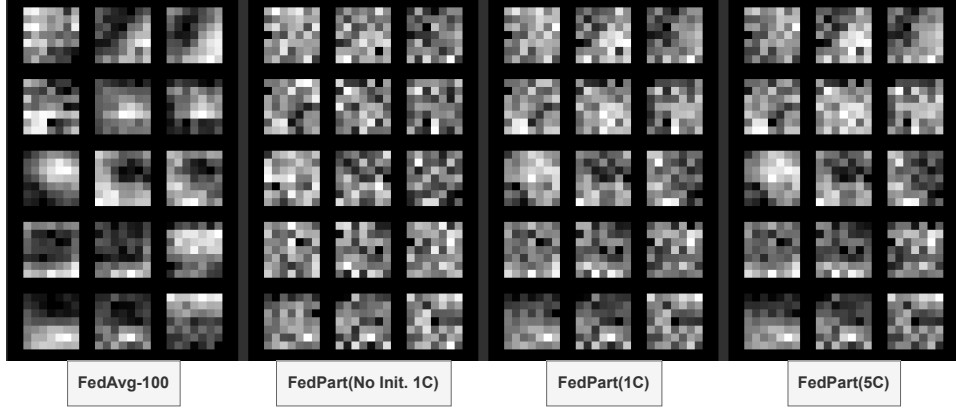

Figure 7: Convolutional kernel visualization results of 5 planes in the first convolutional layer. Each plane include three color channels of image.

# E    Robustness to Privacy Attack

In this section, we will examine the privacy leakage using the DLG method on full network updates and partial network updates. We perform DLG attacks in four settings: transmiting all parameters in FedAvg-100 model; and transmiting only the parameters of layers #1, #9, and #10 separately in FedPart(5C) model. In Fig. 8, we select some representative reconstructed images. The leftmost column represents the original images, while the four columns on the right show the reconstructed images obtained through DLG attacks under different settings.

It can be observed that in the FedAvg-100 scenario, the reconstructed images have the highest quality, exhibiting significant similarity to the original images. However, when adopting partial network updates, the reconstruction quality is poor. Apart from minor color correlations, the reconstructed images exhibit significant differences in structural features compared to the originals. This validates our claim that under the FedPart method, transmitting only a subset of parameters can effectively preserve data privacy.

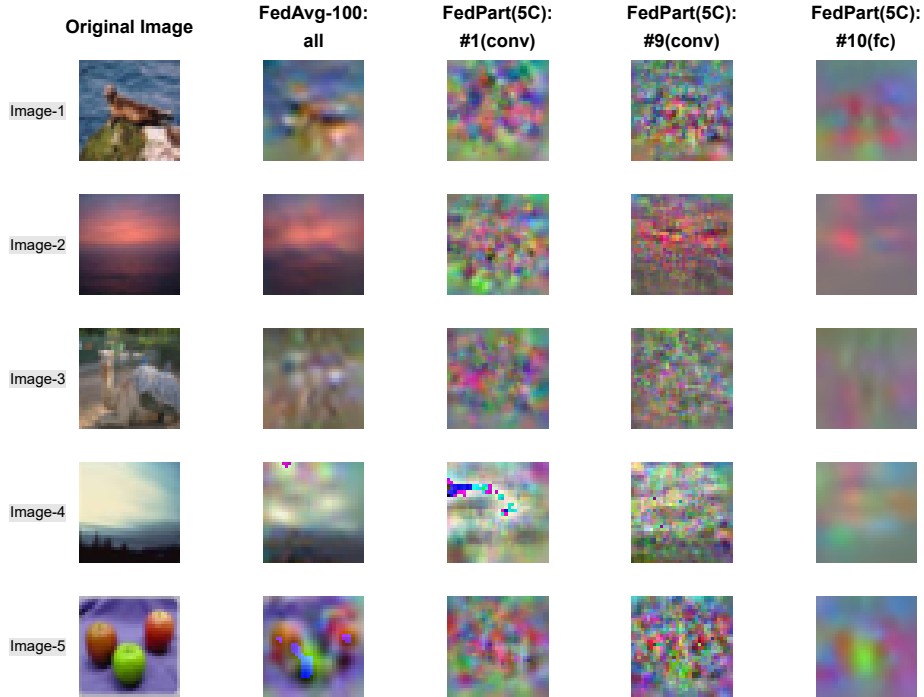

Figure 8: The reconstructed images from DLG attacks on full network of FedAvg-100 and different partial network of FedPart(5C).

# F    Additional Experiments

## F.1    Learning Rate Tuning

In this section, we explore the appropriate learning rate for our experimental configurations. We conduct experiments on the CIFAR-100 dataset using ResNet-8 for both FNU and PNU methods. The experimental results for Adam optimizer with different learning rates are shown in Table 10.

From the results, it can be seen that both FNU and PNU methods perform best with a learning rate of 0.001. So in our experimental configurations, we ultimately select the Adam optimizer with a learning rate of 0.001.

Table 10: Performances (Best Acc.) for different learning rate in full network and partial network updates.

| Dataset | Cycle | FedAvg-FNU | | | FedPart | | |
|---|---|---|---|---|---|---|---|
| | | lr=0.0001 | lr=0.001 | lr=0.01 | lr=0.0001 | lr=0.001 | lr=0.01 |
| CIFAR-100 | 1 | 24.82 | **30.68** | 28.15 | 16.56 | **31.36** | 23.41 |
| | 2 | 30.08 | **32.53** | 30.13 | 21.88 | **34.70** | 30.10 |
| | 3 | 31.91 | **34.02** | 30.93 | 25.42 | **36.34** | 31.94 |
| | 4 | 32.45 | **35.91** | 30.93 | 27.85 | **37.12** | 32.54 |
| | 5 | 32.95 | **35.91** | 31.65 | 29.98 | **37.70** | 33.31 |

## F.2 Evaluation of Client Sampling

In this section, we conduct experiments with 150 clients, randomly sampling 20% of the clients for training and aggregation in each communication round. The experimental results are shown in Table 11. Our method achieves final performance improvements of +2.1%, +1.6%, and +3.4% on CIFAR-10, CIFAR-100, and Tiny-ImageNet, respectively, indicating that FedPart performs better than FedAvg in this scenario.

Table 11: Performance of FedPart with client sampling.

| Dataset | C | FedAvg-FNU | FedPart |
|---|---|---|---|
| CIFAR-10 | 2 | 60.82 | 63.22 |
| | 3 | 61.50 | 63.22 |
| | 4 | 64.34 | 66.08 |
| | 5 | 65.00 | **67.08** |
| CIFAR-100 | 2 | 34.82 | 37.13 |
| | 3 | 39,36 | 37.13 |
| | 4 | 39.64 | 41.00 |
| | 5 | 40.55 | **42.12** |
| Tiny-ImageNet | 2 | 19.63 | 23.06 |
| | 3 | 19.63 | 23.06 |
| | 4 | 22.01 | 26.03 |
| | 5 | 23.33 | **26.75** |

## F.3 Analysis under Extreme Data Heterogeneity

In this section, we conduct experiments with an $\alpha = 0.1$ setting as data heterogeneity is more severe. The experimental results are shown in Table 12.

It can be seen that, in this extreme non-IID scenario ($\alpha = 0.1$), the model accuracy of our method is roughly on par with that of the full parameter method. However, this does not imply that FedPart offers no performance advantages—the benefits primarily arise from reduced communication and computation costs. The results indicate that FedPart can achieve similar accuracy to FedAvg while significantly reducing communication and computation costs (these metrics are consistent with those observed in the IID scenario). As shown in Table 1, when training on Tiny-ImageNet, FedPart reduces communication overhead by 72% and computation overhead by 27%. Therefore, we believe that even in such an extreme scenario of data heterogeneity, our method still holds practical value.

Table 12: Performance of FL algorithms with full network and partial network updates under extreme data heterogeneity (Dirichlet, $\alpha = 0.1$)

| Data | C | FedAvg | | FedProx | |
|---|---|---|---|---|---|
| | | FNU | FedPart | FNU | FedPart |
| CIFAR-10 | 1 | 33.79 | 44.02 | 39.64 | 43.85 |
| | 2 | 44.08 | 44.41 | 46.88 | 45.42 |

# G   Justification of Assumption 3

In Assumption 3 in Section 3.3, we assume that for any mask matrices $S_1, S_2$, it holds that:

$$\frac{\mathbb{E}_{x \sim \mathbf{D_i}}[|||S_1 \odot (\nabla \mathcal{L}(x|w) - \nabla f_i(w))|||]}{\mathbb{E}_{x \sim \mathbf{D_i}}[|||S_2 \odot (\nabla \mathcal{L}(x|w) - \nabla f_i(w))|||]} \leq k, \forall i, w, x \in \mathbf{D_i}, S_1, S_2 \tag{15}$$

Regarding the value of $k$ on the right side of the equation above, recall Eq. 14 in Appendix B, the convergence rate of FedPart satisfies:

$$\frac{1}{T} \sum_{i=1}^{T} \mathbb{E}[||S_i^t \odot \nabla f(\bar{w}^{t-1})||^2 \leq \frac{2}{\gamma T}(f(\bar{w}^0) - f^*) + 4\gamma^2 E^2 G^2 L^2 + \frac{L}{NM}\gamma\sigma^2 k^2$$

Therefore, theoretically, the smaller the value of $k$, the smaller the value on the right side of this inequality, leading to improved convergence of FedPart. Hence, it is important to carefully examine the value range of $k$ in practice.

We begin with analysing the lower bound of $k$. Since $S_1$ and $S_2$ are arbitrary, it is possible that $S_1 = S_2$, indicating a lower bound of 1 for the value $k$. As for approximating the upper bound of the $k$, we conduct Monte Carlo simulations on real-world nueral networks.

We test the $k$ values in three neural networks at different training stages. For each neural network, we conduct Monte Carlo simulations to collect 10,000 samples to accurately approximate the value of $k$. The experimental results are shown in Table 13. We can see that $k$ is close to 1 under different settings, which proves that the effect of applying different masks to the variability of gradient is similar, thus strongly supporting Assumption 3.

Table 13: Monte Carlo simulation experiments for the value of $k$.

|  | ResNet-8 | ResNet-18 |
| --- | --- | --- |
| **0% Training** (Random initialized) | 1.09 | 1.08 |
| **50% Training** (Intermediate) | 1.13 | 1.18 |
| **100% Training** (Fully trained) | 1.13 | 1.17 |

