# OpenReview forum: "Why Go Full? Elevating Federated Learning Through Partial Network Updates"
_NeurIPS.cc/2024/Conference — NeurIPS 2024 poster_

### Official Review · Reviewer_6JAi · 2024-07-04

**Soundness:** 2
**Presentation:** 3
**Contribution:** 2
**Rating:** 5
**Confidence:** 3

**Summary:**

The authors observe 'layer mismatch' phenomenon in federated learning, and propose FedPart that uses partial network updates to address the former issue. They show that FedPart outperforms previous methods and also reduced communication and computation overhead.

**Strengths:**

The proposed idea is neat.

The demonstrated performances in experimental section are (surprisingly) good.

**Weaknesses:**

1. The 'layer mismatch' term does not convince me. The authors try to use Figure 1(a) to show that the 'update stepsize' (to be clarified later) increases after each averaging and thus it indicates layer mismatch, I don't fully understand the logic here, in particular why does this draw the conclusion that it is caused by 'inadequate cooperation among layers'? I do see there is a comparison in Fig1(b), but on one hand it does not differ that much (except for the 50 to 60 iterations region), on the other hand the logic that this is due to 'layer mismatch' is still not perfectly sound even if Fig1(b) shows greater difference. Are there other evidences or maybe related observations in previous literature to support this claim?

2. The proposed strategy to select trainable layers does not provide much insight. Before getting to that section, I was expecting some unique or more informative way to choose what layer to update. It seems like the proposed strategy is the first thing anyone would try, thus I would like to see some argument/support for the effectiveness of choosing this strategy.

3. The experiment setup, if I understand correctly, involves 5 rounds of full network training between each cycle, which I assume is a lot given that partial update is about 8 or 18 rounds per cycle (for Resnet-8 and Resnet-18 respectively)? This is where I concern about the impact of these full network trainings. Since the amount of full network trainings is significant in FedPart, one cannot confidently tell if it is really the partial network trainings that are truly beneficial.

4. Regarding communication cost, I might argue that most communication cost are overhead in each communication, which I believe sharing 1/8 of the number of parameters BUT requiring more communication rounds is not necessarily saving communication cost. And my intuition is that only updating one layer at a time increases the total number of communication rounds?

**Questions:**

1. What exactly is the update stepsize in Fig1? Is it the learning rate? If not, what is it? I might have missed the exact definition of it so I would appreciate if the authors can clarify it here.

2. The comparison in Fig1b really does not look like a big difference to me, for some iterations the partial update actually has a slightly bigger swings? And also I don't understand why Fig1a does not match Fig1b in terms of the blue curve, shouldn't they both be full network update and should be the same?

---

> ### Author Rebuttal · Authors · 2024-08-07
>
> Thank you for the reviewers' comments. Next, we will address each of the issues raised one by one.
>
> **W1 & Q1: What exactly is the update stepsize in Fig1?  ... The authors try to use Figure 1(a) to show that the 'update stepsize' increases after each averaging and thus it indicates layer mismatch, ... why does this draw the conclusion that it is caused by 'inadequate cooperation among layers'?**
>
> We address W1 and Q1 together as they are closely related. In this paper, the step size refers to the magnitude of model parameter changes between iterations. Since the learning rate is fixed in this experiment, this value directly reflects the norm of the gradient. Therefore, during a normal neural network training process, the step size gradually decreases towards zero, leading the model to converge to a fixed point. However, we found that in federated learning, the step size significantly increases after each parameter averaging. This increase in step size suggests that while the global model parameters might be generally robust and contain more knowledge, they cannot cooperate well to complete local tasks immediately (i.e., there is a parameter mismatch). Additionally, because the gradient calculation follows the backpropagation algorithm, the gradient of a parameter in one layer is only directly related to the parameters in the subsequent layers, with no direct correlation among parameters within the same layer. Therefore, we believe this mismatch mainly exists between layers and we refer to this phenomenon as layer mismatch.
>
> **W2: The proposed strategy to select trainable layers does not provide much insight ...**
>
> We appreciate the reviewer's comments. In fact, although our method seems simple and intuitive, it is grounded in significant reasoning. The Sequential Training principle addresses the observation that neural networks converge shallow layers before deeper ones, a trend that is even more pronounced in federated learning [1].  From this perspective, in current federated learning, clients need to train deep networks without the shallow layers being fully converged, resulting in wasted computational power. Our sequential training strategy can effectively address this issue, but can lead to significant degradation in neural network performance. Therefore, we proposed another principle, Multi-round Cycle Training, which is inspired by the classic optimization method BCD (Block Coordinate Descent). By repeating cycles of training multiple times, it effectively alleviates the performance degradation caused by layer-wise training.
>
> In addition, before arriving at this final solution, we tried many different approaches, such as various parameter selection strategies (both randomly, reversely) and optimizer designs, some of which are presented in the ablation studies. Furthermore, while the final method's design is relatively simple, it still involves many details, such as determining the timing for full parameter updates and balancing the length of each cycle with the total number of cycles.
>
> **W4: Regarding communication cost, ... (does) only updating one layer at a time increases the total number of communication rounds?**
>
> We will first address the reviewer's fourth weakness, as we believe the third point will be better addressed after clarifying this one. The reviewer misunderstood our experimental results and our method does NOT increase the required number of communication rounds. In all experimental tables, we ensure the number of communication rounds is identical when comparing full parameter and partial parameter training. For example, in Table 1, the row with C=1 shows the performance of each method at the round when partial network training completes the first cycle, ensuring a fair comparison across all algorithms.
>
> **W3: The experiment setup ... involves 5 rounds of full network training between each cycle, which I assume is a lot ... Since the amount of full network trainings is significant in FedPart, one cannot confidently tell if it is really the partial network trainings that are truly beneficial.**
>
> The reviewer's understanding of the experimental setup is generally correct, but we do not agree with the skepticism. Based on the response to weakness 4 above, our method does not increase the communication rounds. Therefore, from an equivalent perspective, in the process where all rounds involve full parameter training, we modified some rounds to partial parameter training and observed significant performance improvements. Logically, this clearly indicates that partial network updates contributed to the improvement.
>
> **Q2: The comparison in Fig1b really does not look like a big difference to me ... And also I don't understand why Fig1a does not match Fig1b in terms of the blue curve**
>
> Regarding the experimental results in Fig. 1b, the reviewer felt the effect was not significant. We respect the different opinions raised by the reviewer. However, we believe that, at least it is fair to say that FedPart can significantly reduce the degree of layer mismatch in extreme cases and generally maintain the original effect in typical cases, which still supports our claim.
> Concerning the reviewer's comment on the difference in the full parameter training curve, we believe there is some misunderstanding. Indeed, the blue curves in Fig 1a and Fig. 1b are NOT calculated in the same way. As the step size is the total magnitude of model updates, if not all dimensions are updated, this magnitude will naturally decrease, but this is an unfair comparison. To make the comparison as fair as possible, the full parameter training curve in Fig. 1b actually shows the "update magnitude of the same parameters as those updated in partial parameter training," ensuring a fair comparison. We apologize for the confusion and will clarify it in a later version.
>
> [1] Unlocking the Potential of Federated Learning for Deeper Models.

---

> ### Comment · Area_Chair_7xhY · 2024-08-12
>
> Dear Reviewer 6JAi,
>
> Could you please respond with how you think about the authors' response? Please at least indicate that you have read their responses.
>
> Thank you,
> Area chair

---

> ### Comment · Reviewer_6JAi · 2024-08-13
>
> I have read the authors' responses and discussions with other reviewers. I believe the authors have addressed some of my concerns (e.g. W3 W4) and questions, so I will update my score.
> However, I still have some concerns about the vaguely proposed term 'layer mismatch', and the authors' responses do not convince me. I asked if there are other evidences or related observations in previous literature to support the claim, but authors did not answer it. This skepticism remains also due to the lack of explanation about Fig 1(b) where I asked why the difference only seems big for the 50 to 60 iterations region but not else where. Considering that Fig 1(b) seems to be the only experimental support for the claim, not addressing questions regarding it makes the 'layer mismatch' claim unsure to me.

---

> > ### Author Response · Authors · 2024-08-14
> >
> > Dear Reviewer,
> >
> > We greatly appreciate your positive and insightful feedback. As the first to introduce and explore the concept of "layer mismatch," we recognize that there is still room for improvement, especially in developing a deeper understanding of the underlying reasons behind this phenomenon. We are committed to further enhancing our work by strengthening the foundation of our findings with both theoretical and empirical evidence. Thank you once again for your feedback.

---

### Official Review · Reviewer_SwoS · 2024-07-10

**Soundness:** 3
**Presentation:** 4
**Contribution:** 2
**Rating:** 7
**Confidence:** 4

**Summary:**

In this paper, the authors suggest a new approach for the network update step in federated learning. Considering that traditional federated averaging updates and aggregates all parameters at once, leading to a divergence between the global model and the local solution to a client’s specific task, they simply propose to only update parts of the network. They show that the layer mismatch between parameters learned locally and their corresponding global version is largely alleviated through this approach.

[Edit after rebuttal: raised score to accept]

**Strengths:**

This paper is well written and the main idea, which seems well motivated and interesting, is communicated clearly. The experiments are extensive and compare across multiple methods, metrics, and ablations.

**Weaknesses:**

There are two main weaknesses, (1) the suggested layer-wise update means that there is only one (linear) weight matrix that is changed between the global and local models through learning on the local data, which seemingly makes the models much *more* vulnerable to data reconstruction compared to an update of the full model. After all, we only need to reconstruct data from a change in a linear map, rather than a change of a deep, non-linear model. While the authors show that for some specific attacks that were tailored for deep model architectures the approach does fine, a simple attack based on the change in the single layer might be much more effective. At least a discussion on this matter would be important. (2) The motivation and initial results seem convincing, yet the experimental setup is counter-intuitive: you “insert five rounds of full network training between each cycle in […] FedPart”, which means you do more *full* training than partial updates. This seems strange, the motivated layer mismatch should happen consistently in every round. Maybe this is a misunderstanding, a clarification is certainly needed.

There are several other issues raised in the Question below, however, I do believe the paper has value and I will (significantly) raise my score if the concerns are answered.

**Questions:**

Major:

- In Figure 1, the update step size after aggregation seems extremely sharp. I was wondering if the momentum terms (or higher order statistics) of the optimizer were properly reset once the client received the global set of parameters? I briefly checked the submitted source code and it seems the reset of Adam or other optimizers is missing, maybe I overlooked it.
- In Section 3.3 could you please comment on the assumption and put into perspective how realistic they are? A brief discussion on the assumptions would be great – taking them as they are just because they are in the literature feels a bit rough.
-  Regarding the communication costs, how do clients know the mask $S^t$? It seems that this is missing from the analysis.
- In Table 6, the ablation on warm-up rounds, it seems that the larger (60 round) warm-up performs significantly better than what was reported before (in Table 2). How does table 2 look when run with longer warm-up? What is the reason not to do that?
- In the section on the warm-up ablation you mention that FedPart still enhances the model’s accuracy even after so many warm-up rounds (second to last sentence). Where can I see this in the presented numbers?
- According to the paper checklist you provide repetitions to assess experimental statistical significance. This is not given for Table 6-9, which is in my opinion fine for these ablations, but the checklist argument should hence be No. You can add in the description that you assessed the significance for the main experiments with an indication which one these are.

Minor:

- In Figure 2, it is a bit unclear to me why the layers have these different shapes, could you please clarify?
- The activation maximization approach is extremely old and yields arguably useless results. There is more than a decade of advancements in this field, with MACO [1], yielding great visual representations that would be much more useful in accessing the meaning.

[1] https://arxiv.org/abs/2306.06805

**Limitations:**

There is almost no discussion of current limitations of the method at hand. The manuscript would benefit from a slightly more critical discussion in the future.

---

> ### Author Rebuttal · Authors · 2024-08-03
>
> Thank you for the reviewers' comments. Here are our responses.
>
> **Q1: In Figure 1, the update step size ... seems extremely sharp ... if the momentum terms ... were properly reset?**
>
> We did reset the optimizer's state before the start of each local round. The relevant code is at line 131 in the file `fling/component/client/base_client.py`.
>
> **Q2: In Section 3.3 ... please comment on the assumption and put into perspective how realistic they are.**
>
> Thank you for the suggestion. Here are more detailed comments about the assumptions.
>
> Assumptions 1 means that the gradient of the loss function does not change too abruptly as we move in the parameter space. This is reasonable for most models, where the loss changes gradually as the parameters are adjusted. Assumption 2 suggests that the variability in the gradient estimates is controlled. This is generally true in datasets without much extreme values, where individual data points do not drastically affect the overall gradient.
>
> Assumption 3 means that the effect of applying different masks to the variability of gradient should be similar. As this assumption is first proposed in this paper, we next give some evidence. We conducted Monte Carlo simulations (10,000 samples) to approximate the value of $k$ in Assumption 3. As is shown the the table below, $k$ is close to 1 under different settings, supporting Assumption 3.
>
> |           | 0% Training (Random initialized) | 50% Training (Intermediate) | 100% Training (Fully trained) |
> | ------ | ------- | ------ | -------- |
> | ResNet-8  | 1.09                             | 1.13                        | 1.13                          |
> | ResNet-18 | 1.08                             | 1.18                        | 1.17                          |
>
> **Q3: Regarding the communication costs, how do clients know the mask ?**
>
> Actually, the server does not need to transfer the mask $S^t$ to each client. Since the trainable parameters are determined at the *layer* level, the server only need to transfer the *indexes* of trainable layers, resulting in almost no communication overhead. Such a notation is just for easier formulation.
>
> **Q4: ... the larger (60 round) warm-up performs significantly better ... How does table 2 look when run with longer warm-up? What is the reason not to do that?**
>
> *Extending the warm-up period leads to higher final model accuracy but also increases computational and communication overhead*. For instance, to achieve extreme performance, one might need to train with FedAvg until full convergence before adding several rounds of partial network training. The reason why we did not use such a setting in Table 2 is that, we *want to show a balanced improvement for both model accuracy and system overhead*.
>
> **Q5: In the section on the warm-up ablation ... FedPart still enhances the model’s accuracy after so many warm-up rounds. Where can I see this in the presented numbers?**
>
> The relevant figures can be found in Table 6. In the table, *bef.* refers to the accuracy before performing partial network updates (at the end of the warm-up stage), and *aft.* refers to the accuracy after partial parameter training. For example, the accuracy after full parameter training at round 60 in CIFAR-10 is 58.92, while after one cycle of partial parameter training, this number become 66.18.
>
> **Q6: ...repetitions to assess experimental statistical significance is not given for Table 6-9 ...**
>
> We appreciate the reviewer’s suggestions. We will amend the checklist arguments accordingly.
>
> **Q7: In Figure 2, it is a bit unclear to me why the layers have these different shapes**
>
> Sorry for the confusion. In Figure 2, we used different shapes to represent different network parameters. Misalignment of shapes between layers indicates a mismatch, while aligned shapes indicate a match between layers.
>
> **Q8: The activation maximization approach is old ..., with MACO, yielding great visual representations that would be much more useful in accessing the meaning.**
>
> We appreciate the suggestion. Due to time constraints, we cannot provide an improved visualization immediately but will adopt modern methods in future versions.
>
> We next address the concerns raised in the "Weakness" section.
>
> **W1: ... (the proposed method) seemingly makes the models much *more* vulnerable to data reconstruction compared to an update of the full model ... a discussion on this matter would be important.**
>
> Thanks for the comment. Compared to full-parameter training, we believe our proposed method will not increase the risk of privacy leakage. We can abstract  the model training process (for both full and partial parameter training) as a mapping: $ (\Delta w_1, \Delta w_2, ..., \Delta w_n) = f(x) $, where the left hand side denotes the updates to each model parameter, and $ x $ is the training data. From a privacy attack perspective, the goal is to find the best $ x $ such that the updates to $ w $ are *as close as possible* to the actual updates in each dimension. This resembles solving a system of equations, where $ x $ are the unknowns, and each dimension of $ w $ update represents an equation. With partial network training, the unknowns $ x $ remain unchanged compared with full parameter training, but the number of equations decreases (i.e., less information for the attack to follow). Therefore, we believe partial network training leakages less information in general.
>
> **W2: ... the experimental setup is counter-intuitive: you “insert five rounds of full network training between each cycle in FedPart”, which means you do more *full* training than partial updates ...**
>
> There may be a misunderstanding. A "round" refers to one communication between the server and clients, while a "cycle" refers to the entire process of training a partial network from shallow to deep layers, including many rounds. For instance, in ResNet-18 with $R/L=2$, one cycle includes 36 rounds, indicating far more partial-parameter training than full-parameter training.

---

> > ### Comment · Reviewer_SwoS · 2024-08-08
> > **Response rebuttal**
> >
> > Thank you for your response.
> >
> > I appreciate all answers, they clarify many details that strengthen the paper, and I would like to see these additional discussions (esp. on the assumptions and privacy) covered in the final version of the manuscript.
> >
> > I raise my score to accept and wish the authors best of luck for the remaining review process.

---

> ### Author Response · Authors · 2024-08-09
>
> Thank you for your positive feedback.
> We appreciate your helpful comments and we will make sure to include the additional discussions on assumptions and privacy in the final manuscript.

---

### Official Review · Reviewer_9NYP · 2024-07-11

**Soundness:** 2
**Presentation:** 2
**Contribution:** 2
**Rating:** 3
**Confidence:** 4

**Summary:**

This paper discovered the layer mismatch challenge in federated learning due to the full network update. To mitigate this challenge, the authors proposed the FedPart method. Specifically, the FedPart method would ask the clients do full network update in the beginning communication rounds, and then it would ask each client to do only train one layer in each round from the shallowest to the deepest. The experiment results indicate that the proposed method is better than the full network FL baselines.

**Strengths:**

1. The paper proposed a novel observation of layer mismatch due to the full model training in federated learning.

2. The single-layer sequential training is communicationally and computationally efficient to the edge devices.

**Weaknesses:**

1. I suggest the author provide more evidence and discussion of the proposed layer mismatch challenge in the paper. In the current version, there is no experiment to validate the proposed challenge and how this challenge happens during the training.

2. In the experiment part, the paper mainly focused on the iid setup. Even in the ablation study (line 273), the paper adopts alpha equals 1 as the non-iid setup. In most of the FL papers, the experiment would adopt an alpha equal to at least 0.1 to simulate the extreme non-iid cases. As a result, the experiment in the paper could not provide useful support to the authors' argument. Also, the client number is very limited (only 40 clients), and the experiment does not contain any client sampling in the discussion.

3. Even though the paper proposed many related studies, the experiment part contains no other layer-wise FL training baselines. As a result, it is really hard to judge whether the proposed method could stand for the SOTA performance.

4. I noticed that some recent papers also proposed similar layer-wise training methods [1,2]. I suggest the author to include them as baselines to compare with.

1. Zhang, Tuo et al. “TimelyFL: Heterogeneity-aware Asynchronous Federated Learning with Adaptive Partial Training.” 2023 IEEE/CVF Conference on Computer Vision and Pattern Recognition Workshops (CVPRW) (2023): 5064-5073.
2. Lee, Sunwoo et al. “Embracing Federated Learning: Enabling Weak Client Participation via Partial Model Training.” IEEE Transactions on Mobile Computing (2024): n. pag.

**Questions:**

Please see the weakness above.

**Limitations:**

Please see the weakness above.

---

> ### Author Rebuttal · Authors · 2024-08-07
>
> Thank you for the reviewers' comments. Below, we address the concerns raised in the "Weakness" section.
>
> **W1: I suggest the author provide more evidence and discussion of the proposed layer mismatch challenge in the paper. In the current version, there is no experiment to validate the proposed challenge and how this challenge happens during the training.**
>
> We conducted experiments to verify the issue of layer mismatch, as shown in Fig. 1. Here we detail the logic of these experiments. In Fig. 1a, we observed that the step size significantly increases after each model averaging, which is detrimental to the final model convergence. We believe this phenomenon is caused by layer mismatch, and thus we designed a method targeting reducing layer mismatch (i.e., layer-wise training). The results show that after using our method, the step size variation is significantly reduced (as shown in Fig. 1b), and the final performance is improved (as shown in Table 1). Therefore, we believe that layer mismatch does exist, affects model convergence, and consequently impacts federated learning performance, which is the core challenge addressed.
>
> Reviewers might have questions about the relationship between step size and layer mismatch, so we provide a more detailed explanation here: Formally, step size resembles the *norm of the gradient* in each model update. In a normal neural network training process, the step size gradually decreases towards zero during training, leading the model to converge to a fixed point. However, in federated learning, the step size significantly increases after each parameter averaging (Fig. 1a), indicating that the global model parameters do not collaborate well to complete local tasks (i.e., there is a parameter mismatch). Additionally, because the gradient calculation follows the backpropagation algorithm, the gradient of a parameter in one layer is only directly related to the parameters in the subsequent layers, with no direct correlation among parameters within the same layer. Therefore, we believe this mismatch mainly exists between layers (i.e., layer mismatch).
>
> **W2: In the experiment part, the paper mainly focused on the iid setup. Even in the ablation study (line 273), the paper adopts alpha equals 1 as the non-iid setup. In most of the FL papers, the experiment would adopt an alpha equal to at least 0.1 to simulate the extreme non-iid cases. As a result, the experiment in the paper could not provide useful support to the authors' argument. Also, the client number is very limited (only 40 clients), and the experiment does not contain any client sampling in the discussion.**
>
> Thank you for the comments. We conducted additional experiments with 150 clients, randomly sampling 20% of the clients for training and aggregation in each communication round. The results are shown below, indicating that FedPart still performs better than FedAvg in this scenario.
>
> |    Dataset    | Cycle | FedAvg |  FedPart  |
> | :-----------: | :---: | :----: | :-------: |
> |   CIFAR-10    |   1   | 55.18  |   58.95   |
> |               |   2   | 60.82  |   63.22   |
> |               |   3   | 61.50  |   63.22   |
> |               |   4   | 64.34  |   66.08   |
> |               |   5   | 65.00  | **67.08** |
> |   CIFAR-100   |   1   | 30.82  |   31.96   |
> |               |   2   | 34.82  |   37.13   |
> |               |   3   | 39.36  |   37.13   |
> |               |   4   | 39.64  |   41.00   |
> |               |   5   | 40.55  | **42.12** |
> | Tiny-ImageNet |   1   | 17.34  |   17.83   |
> |               |   2   | 19.63  |   23.06   |
> |               |   3   | 19.63  |   23.06   |
> |               |   4   | 22.01  |   26.03   |
> |               |   5   | 23.33  | **26.75** |
>
> We also added experiments with an alpha=0.1 setting as suggested by the reviewer. The results are as follows. It can be seen that partial network training still exhibits much faster convergence compared to full network training and achieves comparable accuracy to full parameter training. Here, it can be seen that in extreme data heterogeneity, we were unable to achieve accuracy improvements because the main issue in this setting is client drift rather than layer mismatch. Since our method is not specifically designed to solve data heterogeneity, in future work, we may introduce other solutions for data heterogeneity to improve performance further.
>
> | Dataset  |  C   | FedAvg (FNU) | FedAvg (FedPart) | FedProx (FNU) | FedProx (FedPart) |
> | :------: | :--: | :----------: | :--------------: | :-----------: | :---------------: |
> | CIFAR-10 |  1   |    33.79     |      44.02       |     39.64     |       43.85       |
> |          |  2   |    44.08     |      44.41       |     46.88     |       45.42       |
>
> **W3 & W4 & W5 & W6: Even though the paper proposed many related studies, the experiment part contains no other layer-wise FL training baselines. As a result, it is really hard to judge whether the proposed method could stand for the SOTA performance. I noticed that some recent papers also proposed similar layer-wise training methods [1,2]. I suggest the author to include them as baselines to compare with.**
>
> We appreciate the reviewers' comments. However, these works cannot be used as benchmarks for comparison, as they are designed for a significantly different scenario from ours. Their purpose in using layer-wise training methods is to enable effective training of networks with different architectures in each client. However, if their methods are applied in our scenario (i.e., the same model is used on all clients, and the computational capacity is consistent), these methods will degrade to full parameter training (i.e., FedAvg, which is compared in our paper). Therefore, we believe comparing with these methods is unnecessary.

---

> > ### Comment · Reviewer_9NYP · 2024-08-12
> >
> > I am not convinced by the author's response on W2.
> >
> > Specifically, in your newly-added results, the performance between FedAvg and FedPart is very similar in the challenging dataset (CIFAR-100 and Tiny-ImageNet), and the performance is under the iid setup (alpha = 1) based on my understanding. Also, the provided non-iid test indicates that the FedPart nearly does not provide a performance upgrade compared to the FedAvg, the performance is so close when C equals 2. In addition, the author does not provide the non-iid test on the CIFAR-100 and Tiny-ImageNet. Based on the newly-added results, it would provide a implication that the proposed FedPart only works well on the easy dataset with iid setup.
> >
> > As a result, I would maintain my score.

---

> ### Author Response · Authors · 2024-08-13
>
> Thank you for the reviewer's time and patience. However, we believe there are still some misunderstandings regarding this issue, so we would like to clarify further:
>
> **Q1: Specifically, in your newly-added results, the performance between FedAvg and FedPart is very similar in the challenging dataset (CIFAR-100 and Tiny-ImageNet), and the performance is under the iid setup (alpha = 1) based on my understanding.**
>
> Thank you for the reviewer’s feedback. In fact, *the performance difference between FedAvg and FedPart on challenging datasets is not similar*. As shown in the results, on CIFAR-10, CIFAR-100, and Tiny-ImageNet, the final accuracy achieved by our method is 67.08%, 42.12%, and 26.75%, respectively, while FedAvg achieves 65.00%, 40.55%, and 23.33%, respectively. This means our method outperforms FedAvg by 2.1%, 1.6%, and 3.4%, respectively. These show that the improvement on more challenging datasets is still substantial, even exceeding the gains on simpler datasets.  Please note that in our first table provided during the rebuttal phase, "C" represents different training time points, and when comparing model performance, the final state (C=5) should be considered.
>
> We summarize the performance improvement of our method over FedAvg in three different scenarios, which is shown in the table below. It can be observed that our method generally shows greater improvements on more challenging datasets, which demonstrates the advantages of our approach.
>
> | Experiment Configurations                    | CIFAR-10 | CIFAR-100 | Tiny-ImageNet |
> | -------------------------------------------- | -------- | --------- | ------------- |
> | I.I.D                                        | +1.9%    | +1.9%     | **+3.7%**     |
> | Non-I.I.D. ($\alpha=1$)                      | +1.0%    | +2.2%     | **+3.4%**     |
> | I.I.D. (client\_num=150, sampling\_rate=0.2) | +2.1%    | +1.6%     | **+3.4%**     |
>
> **Q2: Also, the provided non-iid test indicates that the FedPart nearly does not provide a performance upgrade compared to the FedAvg, the performance is so close when C equals 2. In addition, the author does not provide the non-iid test on the CIFAR-100 and Tiny-ImageNet. Based on the newly-added results, it would provide an implication that the proposed FedPart only works well on the easy dataset with iid setup.**
>
> Thank you for the reviewer’s feedback. We have the following points to clarify:
>
> - First, the reviewer mentioned that the proposed FedPart only works well on the easy dataset with iid setup. For our understanding, the reviewer may take $\alpha = 1$ as iid or nearly iid. However, we would like to clarify that $\alpha=1$ represents a scenario with relatively significant data heterogeneity, and it is widely used in the literature [1,2,3,4]. In contrast, $\alpha = 0.1$ is a *extreme* non-iid setting, as the reviewer noted in the comment. Under the setting of $\alpha=1$, as shown in Table 4 of the main text, our method achieves final performance improvements of +1.0%, +2.2%, and +3.4% on CIFAR-10, CIFAR-100, and Tiny-ImageNet, respectively. These performance gains are not negligible.
> - We agree with the reviewer that, in this extreme non-IID scenario ($\alpha = 0.1$), the model accuracy of our method is roughly on par with that of the full parameter method. However, this does not imply that FedPart offers no performance advantages—the benefits primarily arise from *reduced communication and computation costs*. The results indicate that FedPart can achieve similar accuracy to FedAvg while significantly reducing communication and computation costs (these metrics are consistent with those observed in the IID scenario). As shown in Table 1, when training on Tiny-ImageNet, FedPart reduces communication overhead by 72% and computation overhead by 27%. Therefore, we believe that even in such an extreme scenario of data heterogeneity, our method still holds practical value.
>
> The above is our further response to the reviewer's comments. Thank you very much for the reviewer's patience, and we hope that if there are any further questions, the reviewer can feel free to ask us directly.
>
> [1] Xu, Jian, Xinyi Tong, and Shao-Lun Huang. "Personalized federated learning with feature alignment and classifier collaboration." International Conference on Learning Representations (ICLR). 2023.
>
> [2] Oh, Jaehoon, Sangmook Kim, and Se-Young Yun. "Fedbabu: Towards enhanced representation for federated image classification." International Conference on Learning Representations (ICLR). 2022.
>
> [3] Tan, Yue, et al. "Federated learning from pre-trained models: A contrastive learning approach." Advances in neural information processing systems 35 (2022): 19332-19344.
>
> [4] Zhang, Jianqing, et al. "Fedala: Adaptive local aggregation for personalized federated learning." Proceedings of the AAAI Conference on Artificial Intelligence. Vol. 37. No. 9. 2023.

---

> ### Author Response · Authors · 2024-08-14
>
> Dear Reviewer,
>
> Thank you once more for the time and effort you invested in reviewing our paper, as well as for your insightful follow-up feedback regarding the non-iid setting.  In response to your comments, we have made several clarifications, which we hope address at least some of the concerns you raised. We acknowledge that there is still room for improvement under the extreme non-iid setting (e.g., $\alpha=0.1$). Please let us know if there are any additional questions or if further clarification is needed.

---

### Official Review · Reviewer_ZWNU · 2024-07-13

**Soundness:** 3
**Presentation:** 3
**Contribution:** 3
**Rating:** 7
**Confidence:** 3

**Summary:**

The paper proposes a new and novel method to partially train networks to achieve better training efficiency but also, in some cases, better performance.

**Strengths:**

Training efficiency is an extremely important and timely topic. Given that FL aims to have massive networks to train upon any efficiency gains are exacerbated due to the network size. This work, in my opinion, has the following strengths,

 - The idea is simple however it is executed well and the intuition behind it is sound.
 - The paper can be fully reproduced as the code and datasets are provided.
 - The experiment section is thorough and has sufficient experiment to validate the authors' claims.

**Weaknesses:**

As the authors claimed, while the experiments are sufficient - they are rather limited in terms of dataset sizes given the target applications. Further, synthetic data experiments are missing - I would have personally appreciated to see how FedPart performs in the setting of both IID and non-IID data.

**Questions:**

Based on my comments above, I would like to ask the following questions,

 - How would the method perform on IID and non-IID data?
 - Would it be possible to add some experiments regarding the previously mentioned question?

**Limitations:**

The authors have sufficiently addressed limitations of this work.

---

> ### Author Rebuttal · Authors · 2024-08-07
>
> Thank you for the reviewers' comments. Here are our responses.
>
> **Q1: How would the method perform on IID and non-IID data?**
>
> Thank you for the reviewer's comment. We have conducted additional experiments to enrich our analysis of non-IID data scenarios. We added experiments with an alpha=0.1 setting as data heterogeneity is more severe. The results are as follows. It can be seen that partial network training still exhibits much faster convergence compared to full network training while maintaining the final accuracy. This further shows the robustness of our proposed FedPart under data heterogeneity.
>
> | Dataset  |  C   | FedAvg (FNU) | FedAvg (FedPart) | FedProx (FNU) | FedProx (FedPart) |
> | :------: | :--: | :----------: | :--------------: | :-----------: | :---------------: |
> | CIFAR-10 |  1   |    33.79     |      44.02       |     39.64     |       43.85       |
> |          |  2   |    44.08     |      44.41       |     46.88     |       45.42       |
>
> **W1: ... limited in terms of dataset sizes ...**
>
> We acknowledge your concern regarding the dataset sizes used in our experiments. As we mentioned in the paper, we did not use particularly large datasets to test our method. However, the datasets we used are comparable in size to those used in related works in the field [1, 2], and we believe that these datasets are sufficient to demonstrate the efficacy of FedPart. We will aim to incorporate more extensive datasets in future iterations of our research.
>
> **W2: ... synthetic data experiments are missing ...**
>
> We appreciate your suggestion to include synthetic data experiments. This is indeed a worthwhile addition that can provide more insights into the performance of FedPart in various settings. However, due to time constraints, we cannot immediately incorporate synthetic data experiments. We will seriously consider adding synthetic data experiments in both IID and non-IID settings in future versions of our work.
>
> [1] McMahan, Brendan, et al. "Communication-efficient learning of deep networks from decentralized data." Artificial intelligence and statistics. PMLR, 2017.
> [2] Li, Qinbin, Bingsheng He, and Dawn Song. "Model-contrastive federated learning." Proceedings of the IEEE/CVF conference on computer vision and pattern recognition. 2021.

---

> > ### Comment · Reviewer_ZWNU · 2024-08-09
> > **Read your rebuttal**
> >
> > Thank you for the clarifications provided. Taking in account the rest of the reviews/responses thus far, I will be keeping my score as is. I wish the authors the best of luck with the final decision process.

---

> > > ### Author Response · Authors · 2024-08-09
> > >
> > > Thank you for your thoughtful review and for taking the time to consider our clarifications. We greatly appreciate your positive feedback and helpful comments.

---

### Decision · Program_Chairs · 2024-09-25

**Decision:**

Accept (poster)

**Comment:**

This paper proposes a new approach for federated learning, where only parts of the networks are updated through communication and averaging between clients. The proposed method reduces the layer mismatch between locally learned models and the global model, which is a nice observation as pointed out by reviewers. Also, the experiment results are extensive  and strong, indicating that the proposed method is better than the full network FL baselines under multiple metrics.